# GENERATIVE SCENE GRAPH NETWORKS

**Fei Deng**
Rutgers University
fei.deng@rutgers.edu

**Zhuo Zhi**
University of California, San Diego
zzhi@ucsd.edu

**Donghun Lee**
ETRI
donghun@etri.re.kr

**Sungjin Ahn**
Rutgers University
sjn.ahn@gmail.com

## ABSTRACT

Human perception excels at building compositional hierarchies of parts and objects from unlabeled scenes that help systematic generalization. Yet most work on generative scene modeling either ignores the part-whole relationship or assumes access to predefined part labels. In this paper, we propose Generative Scene Graph Networks (GSGNs), the first deep generative model that learns to discover the primitive parts and infer the part-whole relationship jointly from multi-object scenes without supervision and in an end-to-end trainable way. We formulate GSGN as a variational autoencoder in which the latent representation is a tree-structured probabilistic scene graph. The leaf nodes in the latent tree correspond to primitive parts, and the edges represent the symbolic pose variables required for recursively composing the parts into whole objects and then the full scene. This allows novel objects and scenes to be generated both by sampling from the prior and by manual configuration of the pose variables, as we do with graphics engines. We evaluate GSGN on datasets of scenes containing multiple compositional objects, including a challenging Compositional CLEVR dataset that we have developed. We show that GSGN is able to infer the latent scene graph, generalize out of the training regime, and improve data efficiency in downstream tasks.

## 1 INTRODUCTION

Learning to discover and represent objects purely from observations is at the core of human cognition (Spelke & Kinzler, 2007). Recent advances in unsupervised object-centric representation learning have enabled decomposition of scenes into objects (Greff et al., 2019; Lin et al., 2020b; Locatello et al., 2020), inference and rendering of 3D object models (Chen et al., 2020), and object tracking and future generation (Crawford & Pineau, 2019a; Jiang et al., 2020; Lin et al., 2020a). These neuro-symbolic approaches, where the discreteness of discovered objects provides the symbolic representation, facilitate various desired abilities such as out-of-distribution generalization, relational reasoning, and causal inference.

In this paper, we seek to further discover and represent the structure within objects without supervision. Our motivation is that natural scenes frequently contain compositional objects—objects that are composed of primitive parts. We humans can easily identify the primitive parts and recognize the part-whole relationship. Representing objects as explicit composition of parts is expected to be more efficient, since a vast number of complex objects can often be compositionally explained by a small set of simple primitives. It also allows us to imagine and create meaningful new objects.

A well-established representation for part-whole relationships in computer graphics is called the scene graph (Foley et al., 1996). It is a tree whose leaf nodes store models of primitive parts, and whose edges specify affine transformations that compose parts into objects.

While in computer graphics, the scene graph is manually constructed for rendering, in this paper, we are interested in inferring the scene graph from unlabeled images. To this end, we propose Generative Scene Graph Networks (GSGNs). We formulate this model as a variational autoencoder (Kingma & Welling, 2013; Rezende et al., 2014) whose latent representation is a probabilistic scene

graph. In the latent tree, each node is associated with an appearance variable that summarizes the composition up to the current level, and each edge with a pose variable that parameterizes the affine transformation from the current level to the upper level.

The design of the GSGN decoder follows the rendering process of graphics engines, but with differentiable operations helping the encoder to learn inverse graphics (Tieleman, 2014; Wu et al., 2017; Romaszko et al., 2017; Yao et al., 2018; Deng et al., 2019). As a result, the pose variables inferred by GSGN are interpretable, and the probabilistic scene graph supports symbolic manipulation by configuring the pose variables.

One major challenge is to infer the structure of the scene graph. This involves identifying the parts and grouping them into objects. Notice that unlike objects, parts are often stitched together and thus can have severe occlusion, making it hard to separate them. Existing methods for learning hierarchical scene representations circumvent this challenge by working on single-object scenes (Kosiorek et al., 2019) and also providing predefined or pre-segmented parts (Li et al., 2017; Huang et al., 2020). In contrast, GSGN addresses this challenge directly, and learns to infer the scene graph structure from multi-object scenes without knowledge of individual parts.

Our key observation is that the scene graph has a recursive structure—inferring the structure of the tree should be similar to inferring the structure of its subtrees. Hence, we develop a top-down inference process that first decomposes the scene into objects and then further decomposes each object into its parts. This allows us to reuse existing scene decomposition methods such as SPACE (Lin et al., 2020b) as an inference module shared at each level of the scene graph. However, we find that SPACE has difficulty separating parts that have severe occlusion, possibly due to its complete reliance on bottom-up image features. Therefore, simply applying SPACE for decomposition at each level will lead to suboptimal scene graphs. To alleviate this, GSGN learns a prior over plausible scene graphs that captures typical compositions. During inference, this prior provides top-down information which is combined with bottom-up image features to help reduce ambiguity caused by occlusion.

For evaluation, we develop two datasets of scenes containing multiple compositional 2D and 3D objects, respectively. These can be regarded as compositional versions of Multi-dSprites (Greff et al., 2019) and CLEVR (Johnson et al., 2017), two commonly used datasets for evaluating unsupervised object-level scene decomposition. For example, the compositional 3D objects in our dataset are made up of shapes similar to those in the CLEVR dataset, with variable sizes, colors, and materials. Hence, we name our 3D dataset the Compositional CLEVR dataset.

The contributions of this paper are: (i) we propose the probabilistic scene graph representation that enables unsupervised and end-to-end scene graph inference and compositional scene generation, (ii) we develop and release the Compositional CLEVR dataset to facilitate future research on object compositionality, and (iii) we demonstrate that our model is able to infer the latent scene graph, shows decent generation quality and generalization ability, and improves data efficiency in downstream tasks.

## 2 RELATED WORK

**Object-centric representations.** Our model builds upon a line of recent work on unsupervised object-centric representation learning, which aims to eliminate the need for supervision in structured scene understanding. These methods learn a holistic model capable of decomposing scenes into objects, learning appearance representations for each object, and generating novel scenes—all without supervision and in an end-to-end trainable way. We believe such unsupervised and holistic models are more desirable, albeit more challenging to learn. These models can be categorized into scene-mixture models (Greff et al., 2017; 2019; Burgess et al., 2019; Engelcke et al., 2020; Locatello et al., 2020) and spatial-attention models (Eslami et al., 2016; Crawford & Pineau, 2019b; Lin et al., 2020b; Jiang & Ahn, 2020). Compared to these models, we go a step further by also decomposing objects into parts. We use spatial-attention models as the inference module at each level of the scene graph, because they explicitly provide object positions, unlike scene-mixture models. We combine the inference module with a learned prior to help improve robustness to occlusion. This also allows sampling novel scenes from the prior, which is not possible with spatial-attention models.

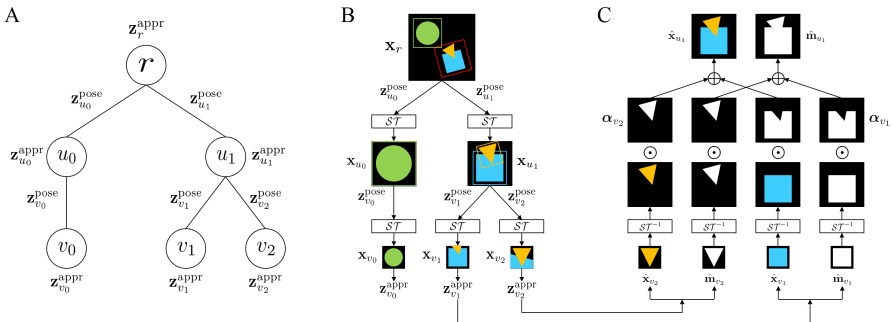

**Figure 1:** *(A)* Probabilistic scene graph representation. Each node represents an entity in the scene, and is associated with an appearance variable. Each edge is associated with a pose variable that specifies the coordinate transformation between the child node and the parent node. *(B)* Spatial attention during inference. We first decompose the scene into high-level objects, and then attend to each object to figure out the constituent parts. *(C)* Recursive decoding process (a single recursive step shown). The image patch $\hat{\mathbf{x}}_{u_1}$ and mask $\hat{\mathbf{m}}_{u_1}$ of an internal node $u_1$ are decoded from the image patches and masks of all its children nodes.

**Hierarchical scene representations.** Modeling the part-whole relationship in scenes has attracted growing interest, and has been utilized for improving image classification (Sabour et al., 2017; Hinton et al., 2018; Kosiorek et al., 2019), parsing, and segmentation (Zhu et al., 2008). However, these models have been applied to scenes with one dominant object only, and cannot perform scene generation. Recent work on assembly-based 3D shape modeling also learns the part-whole relationship (Tulsiani et al., 2017; Li et al., 2017; Zhu et al., 2018; Huang et al., 2020; Kania et al., 2020), but these methods require predefined or pre-segmented parts as input, and can only model single shapes with no background. By contrast, our model learns the part-whole relationship from multi-object scenes without knowledge of individual parts. There has also been work on 3D part decomposition (Chen et al., 2019; Deng et al., 2020), but they require voxels or point clouds as input, and typically focus on geometry (e.g., part occupancy) without learning to represent appearance (e.g., color, material). Part hierarchies have also been used for shape generation (Mo et al., 2019), where the hierarchy is provided as input rather than inferred from the input. Our approach infers compositional structures from static scenes, and is orthogonal to methods that use motion cues for decomposing dynamic scenes (Xu et al., 2019) and methods that infer physical interactions from dynamic scenes (Li et al., 2020; Stanić et al., 2020). Hinton (2021) recently proposed an iterative procedure that is expected to form the part hierarchy through multiple rounds of message passing among adjacent levels. While our model works without iterative message passing, we believe this is important for parsing more complex scenes.

**Hierarchical latent variable models.** Our model can be regarded as a hierarchical latent variable model, and is inspired by several recent advances (Bachman, 2016; Sønderby et al., 2016; Zhao et al., 2017; Maaløe et al., 2019) that have achieved impressive generation quality. While these methods focus on designing the hierarchical structure and training method that harness the full expressive power of generative models, our goal is to learn the hierarchical structure from unlabeled images that captures the compositional relationship among symbolic entities like objects and parts.

## 3 GENERATIVE SCENE GRAPH NETWORKS

### 3.1 GENERATIVE PROCESS

We assume that the image $\mathbf{x}$ is generated by a set of foreground variables collectively denoted $\mathbf{z}_{\text{fg}}$ and a background variable $\mathbf{z}_{\text{bg}}$ as follows:

$$p(\mathbf{x}) = \iint p(\mathbf{x} \,|\, \mathbf{z}_{\text{fg}}, \mathbf{z}_{\text{bg}}) \, p(\mathbf{z}_{\text{bg}} \,|\, \mathbf{z}_{\text{fg}}) \, p(\mathbf{z}_{\text{fg}}) \, \mathrm{d}\mathbf{z}_{\text{fg}} \, \mathrm{d}\mathbf{z}_{\text{bg}} \,. \tag{1}$$

To represent the compositional structures within foreground objects, GSGN models $\mathbf{z}_{\text{fg}}$ as a tree-structured probabilistic scene graph, as shown in Figure 1A. Each leaf node represents a primitive entity that is not further decomposed. Each internal node represents an abstract entity that is composed from its children nodes. Similar to graphics engines, the composition is modeled as affine

transformations, and is specified by the relative pose (including rotation, scaling, and translation) of each child node $v$ with respect to its parent $pa(v)$. We use a pose variable $\mathbf{z}_v^{\text{pose}}$ to represent the relative pose, and associate it with the corresponding edge. We also associate an appearance variable $\mathbf{z}_v^{\text{appr}}$ with each node $v$. It is expected to represent the appearance of entity $v$ in its canonical pose[1], summarizing all lower-level composition in the subtree rooted at $v$. In particular, the appearance variable $\mathbf{z}_r^{\text{appr}}$ at the root node $r$ summarizes the full scene. Due to this summarization assumption, given $\mathbf{z}_v^{\text{appr}}$, we can generate the pose and appearance variables for all children nodes of $v$ in a conditionally independent way. Hence, for a given tree structure with $V$ being the set of all nodes, the prior over foreground variables can be factorized according to the tree structure:

$$p(\mathbf{z}_{\text{fg}}) = p(\mathbf{z}_r^{\text{appr}}) \prod_{v \in V \setminus \{r\}} p(\mathbf{z}_v^{\text{pose}} \,|\, \mathbf{z}_{pa(v)}^{\text{appr}}) \, p(\mathbf{z}_v^{\text{appr}} \,|\, \mathbf{z}_{pa(v)}^{\text{appr}}) \,. \tag{2}$$

Here we further assume conditional independence between $\mathbf{z}_v^{\text{pose}}$ and $\mathbf{z}_v^{\text{appr}}$, since in graphics engines, one should be able to specify the pose and appearance separately.

**Representing tree structures.** The above factorization only works for a given tree structure. To deal with variable tree structures, we need to include them in the latent representation as well. We start by setting a maximum out-degree for each node so that the total number of possible structures is bounded. To determine the structure, it then suffices to specify the presence of each possible edge. Hence, for an arbitrary edge between node $v$ and its parent, we introduce a Bernoulli variable $z_v^{\text{pres}}$ to indicate its presence. If $z_v^{\text{pres}} = 0$, meaning the edge is not present, then the pose variable associated with the edge along with all variables in the subtree rooted at $v$ are excluded from the probabilistic scene graph. More precisely, let us define $\bar{z}_v^{\text{pres}}$ to be the product of all the presence variables along the path from root $r$ to node $v$:

$$\bar{z}_r^{\text{pres}} = 1 \,, \quad \bar{z}_v^{\text{pres}} = z_v^{\text{pres}} \times \bar{z}_{pa(v)}^{\text{pres}} \text{ for } v \in V \setminus \{r\} \,. \tag{3}$$

The foreground variables now become $\mathbf{z}_{\text{fg}} = \{\mathbf{z}_r^{\text{appr}}\} \cup \{z_v^{\text{pres}}, \mathbf{z}_v^{\text{pose}}, \mathbf{z}_v^{\text{appr}}\}_{v \in V \setminus \{r\}}$, and the prior factorizes as follows:

$$p(\mathbf{z}_{\text{fg}}) = p(\mathbf{z}_r^{\text{appr}}) \prod_{v \in V \setminus \{r\}} [p(z_v^{\text{pres}} \,|\, \mathbf{z}_{pa(v)}^{\text{appr}})]^{\bar{z}_{pa(v)}^{\text{pres}}} \, [p(\mathbf{z}_v^{\text{pose}} \,|\, \mathbf{z}_{pa(v)}^{\text{appr}}) \, p(\mathbf{z}_v^{\text{appr}} \,|\, \mathbf{z}_{pa(v)}^{\text{appr}})]^{\bar{z}_v^{\text{pres}}} \,. \tag{4}$$

We implement the pose and appearance variables as Gaussian variables, $p(\mathbf{z}_r^{\text{appr}}) = \mathcal{N}(\mathbf{0}, \mathbf{1})$, and the parameters of each conditional distribution are output by an MLP.

**Differentiable decoder.** We design the decoder to follow the recursive compositing process in graphics engines, helping the encoder to learn inverse graphics. First, for each leaf node $v$, we use a neural network $g(\cdot)$ to decode its appearance variable into a small image patch $\hat{\mathbf{x}}_v$ and a (close to) binary mask $\hat{\mathbf{m}}_v$: $(\hat{\mathbf{x}}_v, \hat{\mathbf{m}}_v) = g(\mathbf{z}_v^{\text{appr}})$. Here, $g(\cdot)$ is implemented as a spatial broadcast decoder (Watters et al., 2019) optionally followed by sub-pixel convolutions (Shi et al., 2016).

We then recursively compose these primitive patches into whole objects and the full scene by applying affine transformations parameterized by the pose variables. Specifically, let $u$ be an internal node, and $ch(u)$ be the set of its children. We compose the higher-level image patch $\hat{\mathbf{x}}_u$ and mask $\hat{\mathbf{m}}_u$ as follows:

$$\hat{\mathbf{x}}_u = \sum_{v \in ch(u)} z_v^{\text{pres}} \cdot \boldsymbol{\alpha}_v \odot \mathcal{ST}^{-1}(\hat{\mathbf{x}}_v, \, \mathbf{z}_v^{\text{pose}}) \,, \tag{5}$$

$$\hat{\mathbf{m}}_u = \sum_{v \in ch(u)} z_v^{\text{pres}} \cdot \boldsymbol{\alpha}_v \odot \mathcal{ST}^{-1}(\hat{\mathbf{m}}_v, \, \mathbf{z}_v^{\text{pose}}) \,. \tag{6}$$

Here, $\odot$ denotes pixel-wise multiplication, and $\mathcal{ST}^{-1}$ is an inverse spatial transformer (Jaderberg et al., 2015) that differentiably places $\hat{\mathbf{x}}_v$ and $\hat{\mathbf{m}}_v$ into the coordinate frame of the parent node $u$. To deal with occlusion, we include relative depth in $\mathbf{z}_v^{\text{pose}}$, and compute a transparency map $\boldsymbol{\alpha}_v$ by the softmax over negative depth values. This ensures that entities with smaller depth will appear in front of entities with larger depth. See Figure 1C for an illustration.

When we reach the root node, we obtain an image $\hat{\mathbf{x}}_r$ and a mask $\hat{\mathbf{m}}_r$ of all foreground objects. We then use a spatial broadcast decoder (Watters et al., 2019) to decode $\mathbf{z}_{\text{bg}}$ into a background image $\hat{\mathbf{x}}_{\text{bg}}$. The full scene $\mathbf{x}$ can now be modeled as a pixel-wise mixture of foreground and background, where $\hat{\mathbf{m}}_r$ serves as the mixing weight:

$$p(\mathbf{x} \,|\, \mathbf{z}_{\text{fg}}, \mathbf{z}_{\text{bg}}) = \hat{\mathbf{m}}_r \odot \mathcal{N}(\mathbf{x} \,|\, \hat{\mathbf{x}}_r, \sigma_{\text{fg}}^2 \mathbf{1}) + (\mathbf{1} - \hat{\mathbf{m}}_r) \odot \mathcal{N}(\mathbf{x} \,|\, \hat{\mathbf{x}}_{\text{bg}}, \sigma_{\text{bg}}^2 \mathbf{1}) \,. \tag{7}$$

Here, $\sigma_{\text{fg}}$ and $\sigma_{\text{bg}}$ are hyperparameters.

---

[1]We use $v$ to refer to both node $v$ in the probabilistic scene graph and the entity that node $v$ represents.

## 3.2 INFERENCE AND LEARNING

Since computing $p(\mathbf{x})$ in Equation 1 involves an intractable integral, we train GSGN with variational inference. The approximate posterior factorizes similarly as the generative process:

$$p(\mathbf{z}_{\text{fg}}, \mathbf{z}_{\text{bg}} \,|\, \mathbf{x}) \approx q(\mathbf{z}_{\text{fg}}, \mathbf{z}_{\text{bg}} \,|\, \mathbf{x}) = q(\mathbf{z}_{\text{fg}} \,|\, \mathbf{x}) \, q(\mathbf{z}_{\text{bg}} \,|\, \mathbf{z}_{\text{fg}}, \mathbf{x}) \,. \tag{8}$$

To infer the foreground variables, we make the key observation that the probabilistic scene graph has a recursive structure. This suggests a recursive inference process, in which the children nodes of the root node are first inferred, and then inference is recursively performed within each subtree rooted at the children nodes. Hence, we design a top-down factorization:

$$q(\mathbf{z}_{\text{fg}} \,|\, \mathbf{x}) = q(\mathbf{z}_r^{\text{appr}} \,|\, \mathbf{x}) \prod_{v \in V \setminus \{r\}} \left[ q(z_v^{\text{pres}} \,|\, \mathbf{z}_{pa(v)}^{\text{appr}}, \mathbf{x}_{pa(v)}) \right]^{\bar{z}_{pa(v)}^{\text{pres}}} \left[ q(\mathbf{z}_v^{\text{pose}} \,|\, \mathbf{z}_{pa(v)}^{\text{appr}}, \mathbf{x}_{pa(v)}) \right]^{\bar{z}_v^{\text{pres}}}$$
$$\times \prod_{v \in V \setminus \{r\}} \left[ q(\mathbf{z}_v^{\text{appr}} \,|\, \mathbf{z}_{pa(v)}^{\text{appr}}, \mathbf{x}_v) \right]^{\bar{z}_v^{\text{pres}}} \,, \tag{9}$$

where the inference of child node $v$ is conditioned on the inferred appearance variable $\mathbf{z}_{pa(v)}^{\text{appr}}$ of its parent. This crucially provides top-down information for separating lower-level entities, since $\mathbf{z}_{pa(v)}^{\text{appr}}$ summarizes lower-level composition. We combine this top-down information with bottom-up image features. Specifically, we use spatial attention to crop a local image region $\mathbf{x}_v$ that is expected to capture the entity $v$. This provides more relevant information for inferring the appearance and structure of entity $v$. The region $\mathbf{x}_v$ is specified by all the predicted pose variables along the path from root $r$ to node $v$. More precisely, we define $\mathbf{x}_r = \mathbf{x}$, and recursively extract $\mathbf{x}_v = \mathcal{ST}(\mathbf{x}_{pa(v)}, \mathbf{z}_v^{\text{pose}})$ using a spatial transformer $\mathcal{ST}$ (Jaderberg et al., 2015), as shown in Figure 1B.

**Parameter sharing between prior and posterior.** Inspired by Ladder VAEs (Sønderby et al., 2016), we find it beneficial to share parameters between the prior and the posterior. Specifically, we implement each factorized posterior as a product of the corresponding prior and a posterior obtained solely from image features. This also allows us to reuse existing scene decomposition methods such as SPACE (Lin et al., 2020b) as an inference module of GSGN. For example,

$$q(\mathbf{z}_v^{\text{appr}} \,|\, \mathbf{z}_{pa(v)}^{\text{appr}}, \mathbf{x}_v) \propto p(\mathbf{z}_v^{\text{appr}} \,|\, \mathbf{z}_{pa(v)}^{\text{appr}}) \, q(\mathbf{z}_v^{\text{appr}} \,|\, \mathbf{x}_v) = p(\mathbf{z}_v^{\text{appr}} \,|\, \mathbf{z}_{pa(v)}^{\text{appr}}) \, q_{\text{SPACE}}(\mathbf{z}_v^{\text{appr}} \,|\, \mathbf{x}_v) \,. \tag{10}$$

**Training.** We use reparameterization trick (Kingma & Welling, 2013; Rezende et al., 2014) to sample the continuous pose and appearance variables, and Gumbel-Softmax trick (Jang et al., 2016; Maddison et al., 2016) to sample the discrete presence variables. The entire model can be trained end-to-end via backpropagation to maximize the evidence lower bound (ELBO):

$$\mathcal{L} = \mathbb{E}_q[p(\mathbf{x} \,|\, \mathbf{z}_{\text{fg}}, \mathbf{z}_{\text{bg}}) - D_{\text{KL}}[q(\mathbf{z}_{\text{fg}} \,|\, \mathbf{x}) \,\|\, p(\mathbf{z}_{\text{fg}})] - D_{\text{KL}}[q(\mathbf{z}_{\text{bg}} \,|\, \mathbf{z}_{\text{fg}}, \mathbf{x}) \,\|\, p(\mathbf{z}_{\text{bg}} \,|\, \mathbf{z}_{\text{fg}})]] \,. \tag{11}$$

**Auxiliary KL loss.** As pointed out by recent work (Chen et al., 2020), learnable priors have difficulty reflecting our prior knowledge. Hence, similar to Chen et al. (2020), we introduce auxiliary KL terms between the posterior and some fixed prior that reflects our preference. In particular, we find it important to introduce a KL term between $q(\mathbf{z}_v^{\text{pres}} \,|\, \mathbf{z}_{pa(v)}^{\text{appr}}, \mathbf{x}_{pa(v)})$ and a Bernoulli prior distribution with low success probability to encourage sparse tree structures.

## 4 EXPERIMENTS

**Datasets.** In order to evaluate GSGN's ability to discover scene graph structures, we develop compositional versions of Multi-dSprites (Greff et al., 2019) and CLEVR (Johnson et al., 2017), two commonly used datasets for evaluating unsupervised object-centric representation learning models. We refer to our new datasets as 2D Shapes and Compositional CLEVR datasets, respectively. For each dataset, we define three types of primitive parts, and ten types of objects composed from these parts. Three of the object types contain a single part, another three contain two parts, and the remaining four contain three parts. To construct a scene, we randomly sample the number of objects (between one and four) and their types, sizes, positions, and orientations. The color and material of each part are also randomly sampled, thus covering a broad range of compositional structures. Each dataset consists of $128 \times 128$ color images, split into 64000 for training, 12800 for validation, and 12800 for testing.

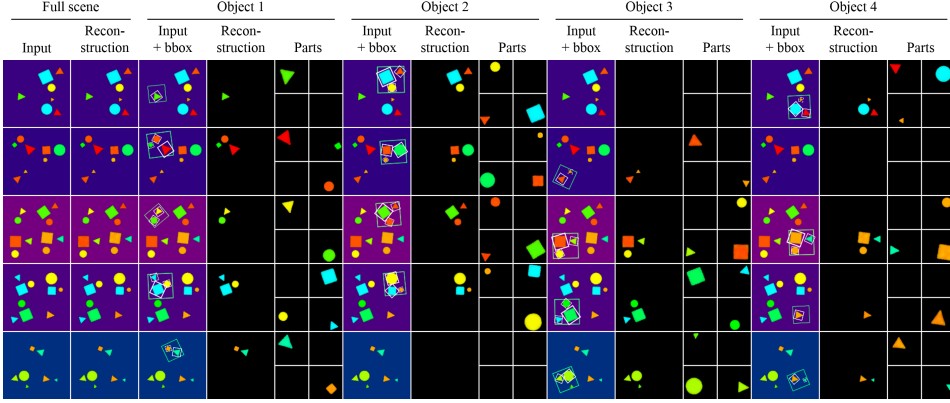

**Figure 2:** Visualization of inferred scene graphs on 2D Shapes dataset.

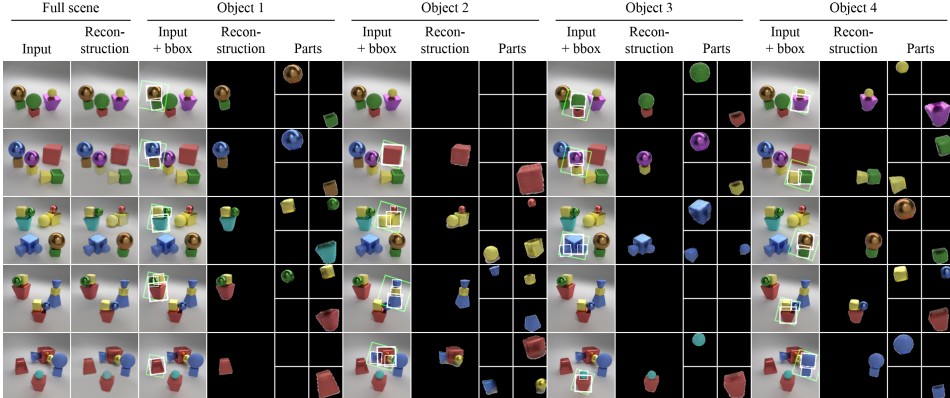

**Figure 3:** Visualization of inferred scene graphs on Compositional CLEVR dataset.

**GSGN implementation.** While our formulation of GSGN allows representing and inferring scene graphs of arbitrary depth, in our experiments, we have only investigated the effectiveness of a three-level GSGN as a proof-of-concept, containing part-, object-, and scene-level representations. We set the maximum out-degree to be 4 for each node. We also implemented another version, denoted GSGN-9, where the maximum out-degree is 9. This is to evaluate the effectiveness of our model when structural hyperparameters are mismatched to data, and show some potential for scalability.

**Baselines.** Previous work on hierarchical scene representations assumes single-object scenes (Kosiorek et al., 2019) and requires predefined or pre-segmented parts (Li et al., 2017; Huang et al., 2020), and thus cannot work on our datasets. Hence, we compare with SPACE (Lin et al., 2020b), a state-of-the-art non-hierarchical scene decomposition model. Although SPACE cannot infer the scene graph, its hyperparameters can be tuned to decompose the scene into either objects (SPACE-O) or parts (SPACE-P). This allows us to compare the inference quality of individual nodes.

**Ablations.** We consider two ablated versions of GSGN. GSGN-No-Share does not share parameters between prior and posterior, namely it does not factorize the posterior as in Equation 10. Instead, the posterior mean and variance are computed by another network that takes in both $\mathbf{z}^{\mathrm{appr}}_{pa(v)}$ and $\mathbf{x}_v$. GSGN-No-Aux directly optimizes the ELBO without auxiliary KL terms.

**Scene graph inference.** We visualize the inferred scene graphs in Figure 2 and Figure 3. Here, the bounding boxes visualize the inferred pose variables (including rotation, scaling, and translation), and the reconstructions are obtained by feeding the inferred appearance variables through the decoder. Empty slots indicate that the corresponding presence variables are zero. As can be seen, GSGN is able to correctly separate the parts and group them into objects, even when objects are close (Figure 2 Row 1-4, Figure 3 Row 1-2), parts have similar color (Figure 2 Row 4-5, Figure 3 Row 3), and there is occlusion between objects and parts (Figure 3 Row 4-5).

**Table 1:** Quantitative results on 2D Shapes dataset.

| Metric | ELBO | Object Count Accuracy | Object F1 Score | Part Count Accuracy | Part F1 Score |
|---|---|---|---|---|---|
| SPACE-O | 3.96 | 99.79% | 99.95% | — | — |
| SPACE-P | 3.95 | — | — | 98.75% | 96.73% |
| GSGN | 3.96 | 99.76% | 99.85% | 99.54% | 99.91% |
| GSGN-No-Share | 3.96 | 99.72% | 99.90% | 99.49% | 99.89% |
| GSGN-No-Aux | 3.96 | 25.00% | 72.75% | 95.79% | 97.34% |

**Table 2:** Quantitative results on Compositional CLEVR dataset.

| Metric | ELBO | Object Count Accuracy | Object F1 Score | Part Count Accuracy | Part F1 Score |
|---|---|---|---|---|---|
| SPACE-O | 3.50 | 98.02% | 99.17% | — | — |
| SPACE-P | 3.48 | — | — | 92.71% | 99.03% |
| GSGN | 3.49 | 98.66% | 99.37% | 98.63% | 99.77% |
| GSGN-9 | 3.49 | 96.27% | 97.21% | 97.73% | 97.59% |
| GSGN-No-Share | 3.50 | 96.21% | 98.26% | 97.09% | 99.38% |
| GSGN-No-Aux | 3.50 | 25.13% | 72.87% | 12.38% | 79.09% |

**Table 3:** Robustness to occlusion on Compositional CLEVR dataset.

| Min Visible Pixels Per Part | <100 | | 100~200 | | >200 | |
|---|---|---|---|---|---|---|
| Metric | Part Count Accuracy | Part Recall | Part Count Accuracy | Part Recall | Part Count Accuracy | Part Recall |
| SPACE-P | 12.24% | 86.03% | 85.66% | 97.95% | 96.11% | 99.48% |
| GSGN | 95.92% | 98.93% | 98.33% | 99.77% | 98.76% | 99.86% |
| GSGN-9 | 89.80% | 97.35% | 96.92% | 97.85% | 98.12% | 97.62% |
| GSGN-No-Share | 85.71% | 96.34% | 96.13% | 99.15% | 97.56% | 99.46% |

We report quantitative results in Table 1 and Table 2. Here, counting accuracy measures the correctness of the number of nodes in the inferred scene graph. F1 score is the harmonic mean of precision and recall. We define a node to be true positive if it is sufficiently close to a groundtruth entity. Hence, F1 score reflects if the inferred nodes indeed capture entities. We measure closeness by the distance between center positions. The distance threshold for objects and parts are 10 and 5 pixels respectively. As can be seen, all models obtain similar reconstruction quality. GSGN-No-Aux fails to infer the scene graph structure. We find that it tends to set all presence variables to be one, resulting in redundant nodes. With auxiliary KL terms, GSGN and GSGN-No-Share are able to infer sparse structures, achieving comparable object-level decomposition performance with SPACE-O.

On the Compositional CLEVR dataset, GSGN achieves better part-level decomposition than SPACE-P. We observe that this is because SPACE-P has difficulty separating parts that have severe occlusion. We further investigate this by splitting the test set into three occlusion levels. Here we measure the amount of occlusion in an image by the minimum number of visible pixels per part. As shown in Table 3, when an image contains a part that has less than 100 visible pixels, SPACE-P performs significantly worse than GSGN. Its low recall suggests that SPACE-P tends to miss occluded parts. We also find GSGN-No-Share performs worse than GSGN in this scenario, indicating that parameter sharing helps better combine top-down information from the prior to reduce ambiguity caused by occlusion. For GSGN-9, although its number of leaf nodes is far more than the maximum number of parts per scene in our dataset, it only shows a slight drop in performance when compared to GSGN, and it is still much better than SPACE-P at identifying parts that have severe occlusion.

The counting accuracy and F1 score evaluate the object- and part-level separately, without taking into account the overall tree structure. To directly measure the quality of the inferred tree structure, we perform a manual inspection on 100 test images from each dataset. GSGN achieves 100% and

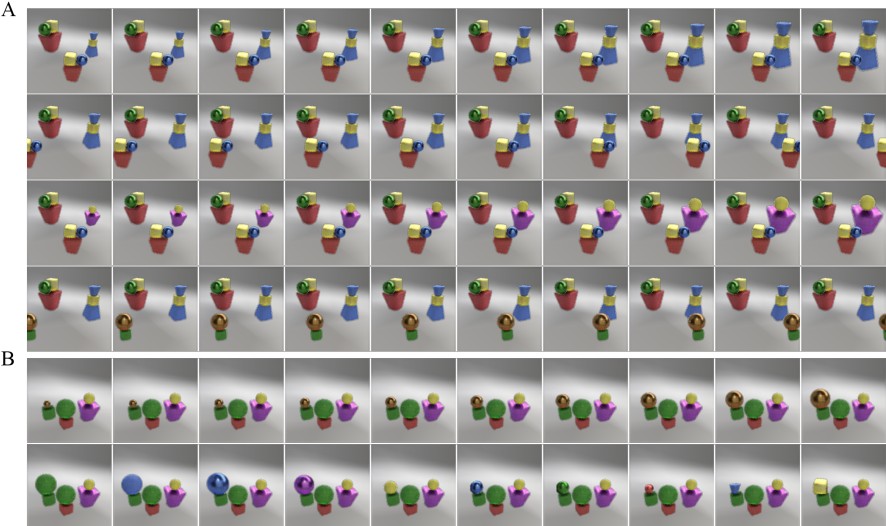

**Figure 4:** Image manipulation results. *(A)* Object-level manipulation. *(B)* Part-level manipulation.

99% structural accuracy on the 2D Shapes and Compositional CLEVR datasets, respectively. We also consider a heuristic baseline, which starts from the parts predicted by SPACE-P, and groups the parts whose bounding boxes overlap. This baseline achieves 95% and 80% structural accuracy on the two datasets, respectively.

**Scene graph manipulation.** The scene graphs inferred by GSGN have interpretable tree structures and pose variables. Therefore, new objects and scenes can be composited by directly manipulating the latent variables in an inferred scene graph. In Figure 4A, we show object-level manipulations. We manipulate the inferred scene graph of the fourth scene of Figure 3, by changing the scale (first row) and $x$-coordinate (second row) of an object in the scene. As can be seen, object occlusion is handled correctly. We can also make these modifications after replacing the object with some other object from a different scene (third and fourth rows). This is achieved by replacing the subtree corresponding to that object. Because the subtree stores the relative pose (rather than the absolute pose) of the parts, the part structure of the newly added object can be automatically preserved. In Figure 4B, we show part-level manipulations on the first scene of Figure 3. In the first row, we modify the size of the bronze ball. This requires changing both the relative position and scale of the ball, so that it remains in contact with the green cube below it. In the second row, we replace the bronze ball with other parts that can be found in the five scenes of Figure 3. This is achieved by replacing the appearance and scale variables with those of the new part, and recalculating the position so that the new part will appear right above the green cube. We note that this allows novel objects to be composited that have never been seen during training (see the last two columns).

**Object and scene generation.** GSGN is able to generate objects and scenes by sampling the latent variables from the learned prior and feeding them through the decoder. We show generation results in Figure 5. We find that GSGN has captured many predefined object types in the two datasets considered, and also managed to come up with novel compositions. The generated scenes are also reasonable, with moderate distance and occlusion between objects. The objects generated by SPACE-O are much less interpretable, since SPACE does not explicitly model the part-whole relationship. Also, SPACE cannot generate meaningful scenes, because the presence variables sampled from its prior will almost always be zero, leading to no object being put onto the scene.

**Generalization performance.** We evaluate GSGN's capacity to generalize to scenes with novel number of objects. We report the metrics in Table 4 and Table 5 (Section A). As can be seen, GSGN demonstrates quite decent generalization performance.

**Data efficiency in downstream tasks.** The compositional structure learned by GSGN can be useful in downstream tasks that require reasoning of part-object relationships. Here we consider a classification task. The input images are generated from the same distribution but with different random

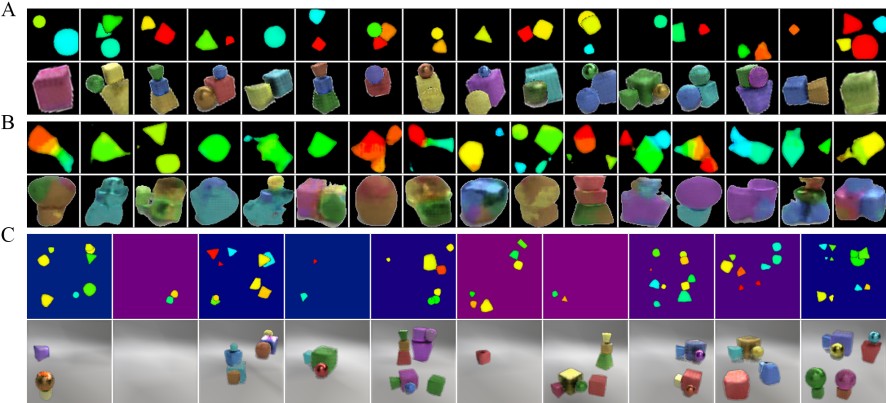

**Figure 5:** *(A)* Objects sampled from GSGN learned prior. *(B)* Objects sampled from SPACE-O prior. *(C)* Scenes sampled from GSGN learned prior.

seeds. The label for each image is obtained by first counting the number of distinct parts within each object, and then summing the count over all objects. Here, two parts are considered the same if they have the same shape, regardless of their pose, color, and material. We expect the representation learned by GSGN to bring better data efficiency compared to SPACE. Notice that SPACE-P cannot group the parts by itself and will fail in this task. Hence, we compare with SPACE-O, whose object-level representation implicitly encodes the constituent parts. We use pretrained GSGN and SPACE-O to obtain the latent representation for each image, and then train a classifier on top of the representation to predict the label. The classifier for SPACE-O first feeds the appearance variable of each object through an MLP, and then performs a learned weighted sum to aggregate them into a scene-level representation vector, which is fed to another MLP to output class probabilities. The classifier for GSGN first aggregates the appearance variables of parts into object-level representation vectors, and then performs the same operations as the classifier for SPACE-O. The tree structure learned by GSGN crucially allows us to use a graph net (Battaglia et al., 2018) within the subtree of each object for aggregation, providing more inductive bias for extracting interactions. We train the two classifiers using 64, 128, 256, 512, 1024, 2048, 4096, 8192, and 16000 training samples. We choose the best learning rate for both classifiers using a fixed validation set of size 12800. We report classification accuracy on a fixed test set also of size 12800. As shown in Figure 6 (Section B), GSGN approximately doubles the data efficiency on this downstream task compared to SPACE-O.

## 5 Conclusion

We have proposed GSGN, the first deep generative model for unsupervised scene graph discovery from multi-object scenes without knowledge of individual parts. GSGN infers the probabilistic scene graph by combining top-down prior and bottom-up image features. This utilizes the higher-level appearance information to guide lower-level decomposition when there is compositional ambiguity caused by severe occlusion. GSGN is also able to generate novel out-of-distribution objects and scenes through scene graph manipulation. While we have demonstrated the effectiveness of a three-level GSGN as a proof-of-concept, it remains an open question whether GSGN can scale to more realistic data, with deeper hierarchies and more complex appearance. Interesting future directions include developing a recurrent module to dynamically control the depth of the scene graph, using iterative inference as suggested by Hinton (2021) for better coordination across levels, and improving the single-level inference module.

### Acknowledgments

This work was supported by Electronics and Telecommunications Research Institute (ETRI) grant funded by the Korean government [21ZR1100, A Study of Hyper-Connected Thinking Internet Technology by autonomous connecting, controlling and evolving ways]. The authors would like to thank Skand Vishwanath Peri, Chang Chen, and Jindong Jiang for helpful discussion, and anonymous reviewers for constructive comments.

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

## A GENERALIZATION PERFORMANCE

**Table 4:** Generalization performance on 2D Shapes dataset.

| Training Set | 1 & 3 Objects | | | | | |
|---|---|---|---|---|---|---|
| Test Set | 2 Objects | | | 4 Objects | | |
| Model | SPACE-O | SPACE-P | GSGN | SPACE-O | SPACE-P | GSGN |
| ELBO | 4.00 | 3.99 | 4.00 | 3.85 | 3.83 | 3.84 |
| Object Count Accuracy | 99.47% | — | 99.51% | 100.0% | — | 100.0% |
| Object F1 Score | 99.89% | — | 99.73% | 100.0% | — | 99.47% |
| Part Count Accuracy | — | 98.12% | 99.21% | — | 96.82% | 99.39% |
| Part F1 Score | — | 95.66% | 99.86% | — | 96.50% | 99.95% |

**Table 5:** Generalization performance on Compositional CLEVR dataset.

| Training Set | 1 & 3 Objects | | | | | |
|---|---|---|---|---|---|---|
| Test Set | 2 Objects | | | 4 Objects | | |
| Model | SPACE-O | SPACE-P | GSGN | SPACE-O | SPACE-P | GSGN |
| ELBO | 3.62 | 3.60 | 3.60 | 3.06 | 3.10 | 3.02 |
| Object Count Accuracy | 98.27% | — | 98.93% | 88.86% | — | 87.78% |
| Object F1 Score | 99.64% | — | 99.72% | 96.72% | — | 96.41% |
| Part Count Accuracy | — | 93.02% | 98.69% | — | 84.96% | 87.20% |
| Part F1 Score | — | 95.66% | 99.71% | — | 98.92% | 97.99% |

## B DATA EFFICIENCY IN DOWNSTREAM TASKS

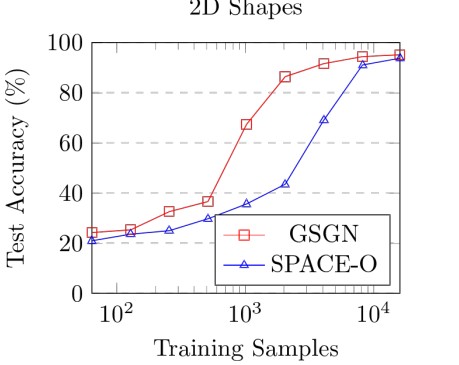
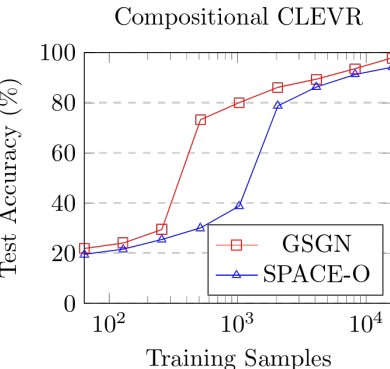

**Figure 6:** Comparison of data efficiency in a downstream classification task that requires reasoning of the part-whole relationship.

## C LEARNING PRIMITIVE TEMPLATES

It is often reasonable to assume that the vast number of complex entities can be composed from only a modest number of primitive entities. Identifying such primitive entities through discrete latent variables would bring additional interpretability. Hence, we consider an extension to GSGN (denoted GSGN-Mem) where the leaf nodes become pointers to a jointly learned memory $\mathbf{M}$ of primitive templates. Each slot in $\mathbf{M}$ stores the low-dimensional shape embedding of a learned primitive part. These embeddings are considered model parameters, and are trained jointly via backpropagation. To link the probabilistic scene graph with the memory of primitives, we decompose the appearance variable $\mathbf{z}_v^{\mathrm{appr}}$ for each leaf node $v$ into $\mathbf{z}_v^{\mathrm{appr}} = (\mathbf{z}_v^{\mathrm{addr}}, \mathbf{z}_v^{\mathrm{what}})$. Here, $\mathbf{z}_v^{\mathrm{addr}}$ is a one-hot addressing

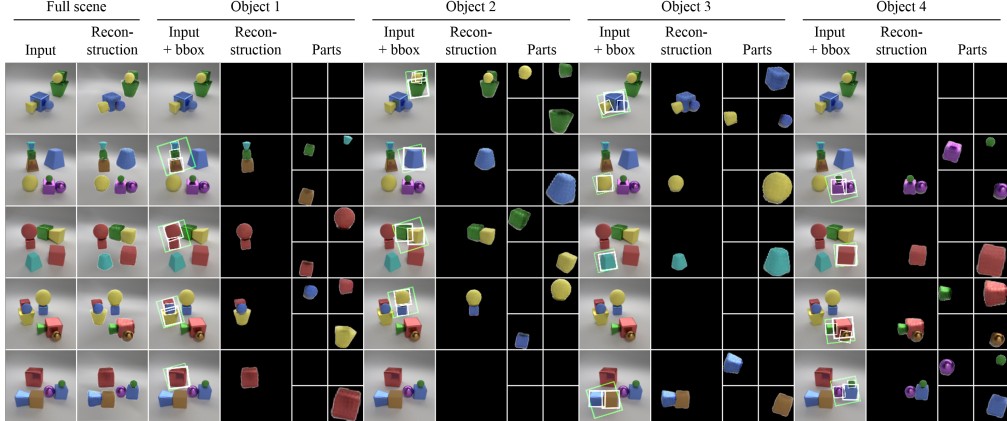

**Figure 7:** Visualization of scene graphs inferred by GSGN-Mem on Compositional CLEVR dataset.

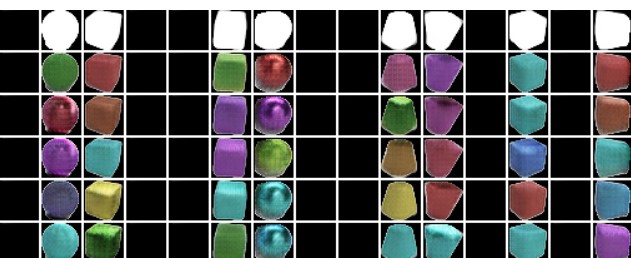

**Figure 8:** Learned memory of primitive parts. Each column corresponds to one slot. The first row shows the decoded mask that captures the shape of the parts. The remaining rows are samples from the learned prior that fill in plausible color, material, and lighting effects.

variable that points to one of the slots in $\mathbf{M}$, and $\mathbf{z}_v^{\text{what}}$ is a continuous variable that explains the remaining variability in the appearance of primitives, such as color, material, and lighting effects. Similar to Bornschein et al. (2017), we factorize the prior as:

$$p(\mathbf{z}_v^{\text{appr}} \,|\, \mathbf{z}_{pa(v)}^{\text{appr}}) = p(\mathbf{z}_v^{\text{addr}} \,|\, \mathbf{z}_{pa(v)}^{\text{appr}}, \mathbf{M}) \, p(\mathbf{z}_v^{\text{what}} \,|\, \mathbf{M}[\mathbf{z}_v^{\text{addr}}]) \,, \tag{12}$$

where $\mathbf{M}[\mathbf{z}_v^{\text{addr}}]$ is the retrieved memory content.

To decode from the templates, we first retrieve the embedding indexed by $\mathbf{z}_v^{\text{addr}}$, and decode it into a (close to) binary mask $\hat{\mathbf{m}}_v = g(\mathbf{M}[\mathbf{z}_v^{\text{addr}}])$ that captures the shape of $v$. We then add color, material, and lighting effects by applying multiplicative modification controlled by $\mathbf{z}_v^{\text{what}}$, and obtain $\hat{\mathbf{x}}_v = \hat{\mathbf{m}}_v \odot h(\mathbf{z}_v^{\text{what}})$. Here, $\odot$ denotes pixel-wise multiplication, $g(\cdot)$ is implemented as a spatial broadcast decoder (Watters et al., 2019), and $h(\cdot)$ a spatial broadcast decoder followed by sub-pixel convolutions (Shi et al., 2016).

We trained GSGN-Mem on the Compositional CLEVR dataset. We find it is able infer the scene graph, as we show qualitatively in Figure 7. The primitives have learned the appearance of all the parts in the dataset, while reserving some memory capacity for potential future use (see Figure 8).

## D IMPLEMENTATION DETAILS

### D.1 FOREGROUND INFERENCE

We provide the implementation outline for computing $q(\mathbf{z}_{\text{fg}} \,|\, \mathbf{x})$ in Algorithm 1 and Algorithm 2. The inference is carried out in a top-down fashion, starting at level 1 that corresponds to the root node. The core inference module, $\text{GSGN}^{(l+1)}$, takes as input the appearance variable $\mathbf{z}_v^{\text{appr}}$ and the local image region $\mathbf{x}_v$ for a node $v$ in level $l$, and outputs the latent variables $\mathbf{z}_{ch(v)}$ and local image regions $\mathbf{x}_{ch(v)}$ for all children nodes of $v$. Here, $ch(v)$ denotes the set of all children nodes of $v$, and

$\mathbf{z}_{ch(v)}$ and $\mathbf{x}_{ch(v)}$ are defined as:

$$\mathbf{z}_{ch(v)} = \{(\mathbf{z}_u^{\text{pres}}, \mathbf{z}_u^{\text{pose}}, \mathbf{z}_u^{\text{appr}})\}_{u \in ch(v)}, \quad \mathbf{x}_{ch(v)} = \{\mathbf{x}_u\}_{u \in ch(v)} . \tag{13}$$

$\text{GSGN}^{(l+1)}$ is shared for all nodes $v$ in level $l$, and serves as a generic module for decomposing an entity at level $l$ into its parts at level $(l+1)$.

There are three submodules in $\text{GSGN}^{(l+1)}$, namely $\text{PresPoseApprPri}^{(l+1)}$ for computing the conditional prior, and $\text{PresPoseEnc}^{(l+1)}$ and $\text{ApprEnc}^{(l+1)}$ that are adapted from the SPACE encoder. All three submodules can be implemented as CNNs. For $\text{PresPoseApprPri}^{(l+1)}$ and $\text{PresPoseEnc}^{(l+1)}$, each cell in the output feature map corresponds to one child node, and the number of cells determines the maximum out-degree at level $l$. Hence, the latent variables at the children nodes can be inferred in parallel through a feedforward pass of each submodule.

---

**Algorithm 1** Foreground Inference

---

**Input:** image $\mathbf{x}$, number of scene graph levels $L$
1: Infer the appearance variable for the root node:

$$q(\mathbf{z}_r^{\text{appr}} \mid \mathbf{x}) = \text{ApprEnc}^{(l=1)}(\mathbf{x})$$
$$\mathbf{z}_r^{\text{appr}} \sim q(\mathbf{z}_r^{\text{appr}} \mid \mathbf{x}), \; \mathbf{x}_r = \mathbf{x}$$

2: **for each** scene graph level $l = 1, 2, \ldots, L - 1$ **do**
3:     **for each** node $v$ in level $l$ **parallel do**
4:         Infer the latent variables and crop local image regions for all children nodes of $v$:

$$\mathbf{z}_{ch(v)}, \mathbf{x}_{ch(v)} = \text{GSGN}^{(l+1)}(\mathbf{z}_v^{\text{appr}}, \mathbf{x}_v)$$

5:     **end for**
6: **end for**
7: **return** all foreground variables $\mathbf{z}_{\text{fg}}$

---

**Algorithm 2** $\text{GSGN}^{(l+1)}$: Foreground Inference at Level $(l + 1)$

---

**Input:** appearance variable $\mathbf{z}_v^{\text{appr}}$ and local image region $\mathbf{x}_v$ for a node $v$ in level $l$
1: Compute conditional prior:

$$p(\mathbf{z}_{ch(v)} \mid \mathbf{z}_v^{\text{appr}}) = \text{PresPoseApprPri}^{(l+1)}(\mathbf{z}_v^{\text{appr}})$$

2: Use SPACE to predict presence and pose variables from $\mathbf{x}_v$:

$$q_{\text{SPACE}}(\mathbf{z}_{ch(v)}^{\text{pres,pose}} \mid \mathbf{x}_v) = \text{PresPoseEnc}^{(l+1)}(\mathbf{x}_v)$$

3: Sample the presence and pose variables:

$$\mathbf{z}_{ch(v)}^{\text{pres,pose}} \sim q(\mathbf{z}_{ch(v)}^{\text{pres,pose}} \mid \mathbf{z}_v^{\text{appr}}, \mathbf{x}_v) \propto p(\mathbf{z}_{ch(v)}^{\text{pres,pose}} \mid \mathbf{z}_v^{\text{appr}}) \, q_{\text{SPACE}}(\mathbf{z}_{ch(v)}^{\text{pres,pose}} \mid \mathbf{x}_v)$$

4: **for each** child node $u \in ch(v)$ **parallel do**
5:     Use spatial transformer to crop local image region for node $u$:

$$\mathbf{x}_u = \mathcal{ST}(\mathbf{x}_v, \mathbf{z}_u^{\text{pose}})$$

6:     Use SPACE to predict appearance variable from $\mathbf{x}_u$:

$$q_{\text{SPACE}}(\mathbf{z}_u^{\text{appr}} \mid \mathbf{x}_u) = \text{ApprEnc}^{(l+1)}(\mathbf{x}_u)$$

7:     Sample the appearance variable:

$$\mathbf{z}_u^{\text{appr}} \sim q(\mathbf{z}_u^{\text{appr}} \mid \mathbf{z}_v^{\text{appr}}, \mathbf{x}_u) \propto p(\mathbf{z}_u^{\text{appr}} \mid \mathbf{z}_v^{\text{appr}}) \, q_{\text{SPACE}}(\mathbf{z}_u^{\text{appr}} \mid \mathbf{x}_u)$$

8: **end for**
9: **return** latent variables $\mathbf{z}_{ch(v)}$ and local image regions $\mathbf{x}_{ch(v)}$ for all children nodes of $v$

---

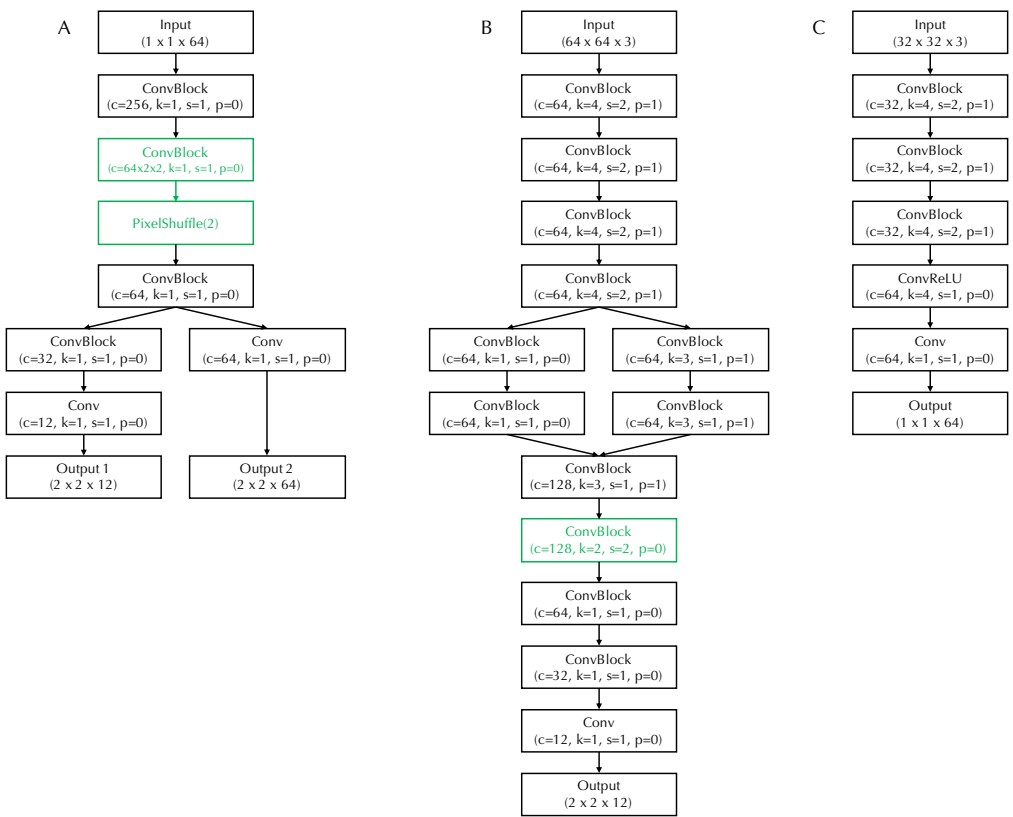

**Figure 9:** CNN architectures of *(A)* PresPoseApprPri$^{(3)}$, *(B)* PresPoseEnc$^{(3)}$, and *(C)* ApprEnc$^{(3)}$ used in our three-level GSGN. To implement GSGN-9, in which the maximum out-degree is 9, we just need to change the layers in green so that the output feature maps have spatial size $3 \times 3$.

For concreteness, in Figure 9, we provide the CNN architectures of the submodules of GSGN$^{(3)}$ used in our experiments. They are responsible for decomposing objects into parts. The submodules at the other levels are similar. In Figure 9, each ConvBlock involves a convolution layer followed by group normalization (Wu & He, 2018) and ReLU nonlinearity (Nair & Hinton, 2010). ConvReLU denotes a convolution layer directly followed by ReLU nonlinearity. The PixelShuffle($r$) layer (Shi et al., 2016) converts an input feature map of size $[H, W, C \times r \times r]$ to an output feature map of size $[H \times r, W \times r, C]$.

PresPoseApprPri$^{(3)}$ (Figure 9A) takes in the 64-dimensional appearance variable of an object, and outputs the conditional prior for each of its four possible children. Here, output 1 includes two (unnormalized) logits for the presence variable, and the mean and standard deviation of the 5-dimensional pose variable, including 2-dimensional position and 1-dimensional rotation, scaling, and depth values. Output 2 includes the mean and standard deviation of the 32-dimensional appearance variable of a part. Notice that due to the PixelShuffle layer, not all parameters are shared across the four children. This allows the network to predict a different distribution for the latent variables of each child. PresPoseEnc$^{(3)}$ (Figure 9B) takes in the cropped image region of an object (resized to $64 \times 64$), and infers the presence and pose of its children. ApprEnc$^{(3)}$ (Figure 9C) takes in the image region of a part (cropped from the input to PresPoseEnc$^{(3)}$ and resized to $32 \times 32$), and infers the appearance of that part. Notice that the cropping and appearance inference can be done in parallel for all parts. We apply Softplus nonlinearity to ensure that the standard deviations are positive.

## D.2 Parameterization of Pose Variables

As mentioned in Section D.1, the pose variable includes position, scaling, rotation, and depth:

$$\mathbf{z}_v^{\text{pose}} = (\mathbf{z}_v^{\text{where}}, \mathbf{z}_v^{\text{scale}}, \mathbf{z}_v^{\text{theta}}, \mathbf{z}_v^{\text{depth}}) \,. \tag{14}$$

These are all Gaussian variables. Before feeding them to the spatial transformer, we first apply nonlinear squashing functions to convert their values into the desired range. In the following, we describe each variable in detail.

**Position.** $\mathbf{z}_v^{\text{where}} \in \mathbb{R}^2$ represents the center position of $v$ in the image patch of its parent $\mathbf{x}_{pa(v)}$. Similar to SPAIR (Crawford & Pineau, 2019b) and SPACE (Lin et al., 2020b), we divide $\mathbf{x}_{pa(v)}$ into $r \times r$ grid cells, with $r^2$ being the maximum out-degree. Each $v$ of the same parent is associated with a cell index $(i, j)$, and is expected to identify an entity in $\mathbf{x}_{pa(v)}$ that is close to cell $(i, j)$. Specifically, $\mathbf{z}_v^{\text{where}}$ is used to compute the deviation of entity $v$ from the center of cell $(i, j)$. Let $\boldsymbol{c}_{i,j} \in \mathbb{R}^2$ denote the center of cell $(i, j)$, and $d \in \mathbb{R}$ denote the side length of each cell. The center position of entity $v$ can be computed as:

$$\tilde{\mathbf{z}}_v^{\text{where}} = \boldsymbol{c}_{i,j} + a \cdot d \cdot \tanh(\mathbf{z}_v^{\text{where}}) \,. \tag{15}$$

Here, $a$ is a hyperparameter. If $a = 0.5$, then the center of entity $v$ must be within cell $(i, j)$. Typically, $a$ is set to be greater than 0.5 so that the network can learn to distribute the entities that are close together across multiple cells. In our experiments, we set $a = 0.55$ for GSGN ($r = 2$), and $a = 0.75$ for GSGN-9 ($r = 3$).

**Scaling.** $\mathbf{z}_v^{\text{scale}}$ represents the scale of $v$ in $\mathbf{x}_{pa(v)}$. If $v$ is an object, then $\mathbf{z}_v^{\text{scale}} \in \mathbb{R}^2$, and the object scale is computed as:

$$\tilde{\mathbf{z}}_v^{\text{scale}} = 0.3 + 0.29 \cdot \tanh(\mathbf{z}_v^{\text{scale}}) \,. \tag{16}$$

If $v$ is a part, then for simplicity we let $\mathbf{z}_v^{\text{scale}} \in \mathbb{R}$, and compute the part scale as:

$$\tilde{\mathbf{z}}_v^{\text{scale}} = 0.5 + 0.49 \cdot \tanh(\mathbf{z}_v^{\text{scale}}) \,. \tag{17}$$

Here, $\tilde{\mathbf{z}}_v^{\text{scale}} = \mathbf{1}$ would mean that entity $v$ has the same size as its parent. We find that limiting the range of object scale is necessary for GSGN-No-Aux to learn object-level decomposition (otherwise, GSGN-No-Aux tends to explain the entire scene as one object). With the auxiliary KL terms introducing some preferred prior (described in Section D.3), in GSGN the object scale can probably be computed in the same way as the part scale, but we have not investigated this.

**Rotation.** $\mathbf{z}_v^{\text{theta}} \in \mathbb{R}$ represents the rotation of $v$ in $\mathbf{x}_{pa(v)}$. Since there is no groundtruth, the network is free to decide what each entity should look like when there is no rotation, and predict the rotation angles accordingly. On the 2D Shapes dataset, the rotation is computed as:

$$\tilde{\mathbf{z}}_v^{\text{theta}} = \pi \cdot \tanh(\mathbf{z}_v^{\text{theta}}) \,. \tag{18}$$

On the Compositional CLEVR dataset, since all objects are placed upright on a plane, we compute the rotation as:

$$\tilde{\mathbf{z}}_v^{\text{theta}} = \frac{\pi}{10} \cdot \tanh(\mathbf{z}_v^{\text{theta}}) \,. \tag{19}$$

**Depth.** $\mathbf{z}_v^{\text{depth}} \in \mathbb{R}$ represents the relative depth of $v$ with respect to its siblings (other children of $pa(v)$). It is transformed to a positive value:

$$\tilde{\mathbf{z}}_v^{\text{depth}} = \text{Softplus}(\mathbf{z}_v^{\text{depth}}) \,. \tag{20}$$

$\tilde{\mathbf{z}}_v^{\text{depth}}$ is then used to construct a transparency map $\boldsymbol{\alpha}_v$ that is the same size as $\mathbf{x}_{pa(v)}$. Ideally, for a pixel $(i, j)$, $\boldsymbol{\alpha}_v(i, j) = 1$ if entity $v$ contains this pixel and at the same time has a smaller $\tilde{\mathbf{z}}_v^{\text{depth}}$ than all its siblings that also contain the pixel, and $\boldsymbol{\alpha}_v(i, j) = 0$ otherwise. To determine which pixels are contained in entity $v$, we can simply use the decoded mask $\hat{\mathbf{m}}_v$ and the presence variable $\mathbf{z}_v^{\text{pres}}$. Specifically, we first place the mask into the parent coordinate frame by an inverse spatial transformer:

$$\tilde{\mathbf{m}}_v = \mathbf{z}_v^{\text{pres}} \cdot \mathcal{ST}^{-1}(\hat{\mathbf{m}}_v, \, \tilde{\mathbf{z}}_v^{\text{where}}, \, \tilde{\mathbf{z}}_v^{\text{scale}}, \, \tilde{\mathbf{z}}_v^{\text{theta}}) \,, \tag{21}$$

where $\tilde{\mathbf{m}}_v$ is the same size as $\mathbf{x}_{pa(v)}$, and ideally $\tilde{\mathbf{m}}_v(i, j) = 1$ if entity $v$ is present and contains pixel $(i, j)$, and $\tilde{\mathbf{m}}_v(i, j) = 0$ otherwise (in practice, $\boldsymbol{\alpha}_v$ and $\tilde{\mathbf{m}}_v$ are not strictly binary). We can now compute $\boldsymbol{\alpha}_v(i, j)$ by a masked Softmax over negative depth values:

$$\boldsymbol{\alpha}_v(i, j) = \frac{\tilde{\mathbf{m}}_v(i, j) \cdot \exp\left(-\tilde{\mathbf{z}}_v^{\text{depth}}\right)}{\sum_{u \in ch(pa(v))} \tilde{\mathbf{m}}_u(i, j) \cdot \exp\left(-\tilde{\mathbf{z}}_u^{\text{depth}}\right)} \,. \tag{22}$$

### D.3 AUXILIARY KL TERMS

In this section, we list the auxiliary KL terms used in our experiments. They have the same weight as the original KL terms that appear in the ELBO.

Table 6 lists the auxiliary KL terms that are the same across all levels. We use a Bernoulli prior with low success probability for $z_v^{\text{pres}}$ to encourage sparse tree structures. The prior for $z_v^{\text{depth}}$ has its mean value set to $4$. This helps avoid the negative half of Softplus nonlinearity that tends to make the transformed $\tilde{z}_v^{\text{depth}}$ values (Equation 20) less distinguishable from each other.

Table 7 lists the auxiliary KL terms that are only used at the object level for the Compositional CLEVR dataset. The prior for $\mathbf{z}_v^{\text{scale}}$ has its mean value set to $-0.2$. According to Equation 16, this corresponds to an approximate mean value of $0.24$ for object size $\tilde{z}_v^{\text{scale}}$. We note that this value does not match the average object size in the dataset. It is simply set to be small to encourage decomposition and avoid explaining the entire scene as one object. The prior for $\mathbf{z}_v^{\text{appr}}$ is used only in GSGN-9. We find that it helps prevent posterior collapse, which we observed in GSGN-9 but not in GSGN. On the 2D Shapes dataset, we use a Gaussian mixture prior for $\mathbf{z}_v^{\text{scale}}$, as shown in Table 8. This is to reflect the prior knowledge that these 2D objects can have significantly different scales. Neither the mean values nor the mixing weights are chosen to match the actual distribution of object scales in the dataset. We estimate this auxiliary KL term using 100 samples from the posterior. This is not expensive since $\mathbf{z}_v^{\text{scale}}$ has only two dimensions.

Finally, as shown in Table 9, we use a Gaussian mixture prior for the part scales. The component mean values, after being transformed according to Equation 17, are approximately 0.91, 0.5, and 0.24. Again, we note that neither the mean values nor the mixing weights match the actual distribution of part scales in our datasets. These values are chosen to encourage decomposition of objects into parts and at the same time reflect the prior knowledge that parts can have significantly different scales. We estimate this auxiliary KL term using 100 samples from the posterior.

**Table 6:** Auxiliary KL terms used at all levels.

| Posterior | Unconditioned Prior |
|-----------|---------------------|
| $q(z_v^{\text{pres}} \mid \mathbf{z}_{pa(v)}^{\text{appr}}, \mathbf{x}_{pa(v)})$ | $\text{Bernoulli}(1 \times 10^{-10})$ |
| $q(\mathbf{z}_v^{\text{where}} \mid \mathbf{z}_{pa(v)}^{\text{appr}}, \mathbf{x}_{pa(v)})$ | $\mathcal{N}(\mathbf{0}, \mathbf{1})$ |
| $q(z_v^{\text{theta}} \mid \mathbf{z}_{pa(v)}^{\text{appr}}, \mathbf{x}_{pa(v)})$ | $\mathcal{N}(0, 1)$ |
| $q(z_v^{\text{depth}} \mid \mathbf{z}_{pa(v)}^{\text{appr}}, \mathbf{x}_{pa(v)})$ | $\mathcal{N}(4, 1)$ |

**Table 7:** Auxiliary KL terms used only at the object level on Compositional CLEVR dataset.

| Posterior | Unconditioned Prior |
|-----------|---------------------|
| $q(\mathbf{z}_v^{\text{scale}} \mid \mathbf{z}_{pa(v)}^{\text{appr}}, \mathbf{x}_{pa(v)})$ | $\mathcal{N}(-\mathbf{0.2}, \mathbf{0.1})$ |
| $q(\mathbf{z}_v^{\text{appr}} \mid \mathbf{z}_{pa(v)}^{\text{appr}}, \mathbf{x}_v)$ | $\mathcal{N}(\mathbf{0}, \mathbf{1})$ |

**Table 8:** Auxiliary KL term used only at the object level on 2D Shapes dataset.

| Posterior | Unconditioned Prior |
|-----------|---------------------|
| $q(\mathbf{z}_v^{\text{scale}} \mid \mathbf{z}_{pa(v)}^{\text{appr}}, \mathbf{x}_{pa(v)})$ | $0.2\mathcal{N}(-\mathbf{0.2}, \mathbf{0.1}) + 0.8\mathcal{N}(-\mathbf{1.2}, \mathbf{0.1})$ |

**Table 9:** Auxiliary KL term used only at the part level.

| Posterior | Unconditioned Prior |
|---|---|
| $q(z_v^{\text{scale}} \mid \mathbf{z}_{pa(v)}^{\text{appr}}, \mathbf{x}_{pa(v)})$ | $0.1\mathcal{N}(1.2, 0.1) + 0.1\mathcal{N}(0, 0.05) + 0.8\mathcal{N}(-0.6, 0.1)$ |

### D.4 Background inference

The prior and posterior for the background variable are implemented as:

$$p(\mathbf{z}_{\text{bg}} \mid \mathbf{z}_{\text{fg}}) = p(\mathbf{z}_{\text{bg}} \mid \mathbf{z}_r^{\text{appr}}) , \quad q(\mathbf{z}_{\text{bg}} \mid \mathbf{z}_{\text{fg}}, \mathbf{x}) = q(\mathbf{z}_{\text{bg}} \mid \mathbf{z}_r^{\text{appr}}, \mathbf{x}) . \quad (23)$$

We provide the implementation outline for computing $q(\mathbf{z}_{\text{bg}} \mid \mathbf{z}_r^{\text{appr}}, \mathbf{x})$ in Algorithm 3. The two submodules, BgPri and BgEnc, are implemented as CNNs, and their architectures can be found in Figure 10. Both submodules output the mean and standard deviation of the background variable. We apply Softplus nonlinearity to ensure that the standard deviations are positive. $\mathbf{z}_r^{\text{appr}}$ and $\mathbf{z}_{\text{bg}}$ have dimensions 128 and 8, respectively.

---

**Algorithm 3** Background Inference

**Input:** image $\mathbf{x}$, appearance variable of the root node $\mathbf{z}_r^{\text{appr}}$

1: Compute conditional prior:

$$p(\mathbf{z}_{\text{bg}} \mid \mathbf{z}_r^{\text{appr}}) = \text{BgPri}(\mathbf{z}_r^{\text{appr}})$$

2: Predict the background variable from $\mathbf{x}$:

$$q_{\text{SPACE}}(\mathbf{z}_{\text{bg}} \mid \mathbf{x}) = \text{BgEnc}(\mathbf{x})$$

3: Sample the background variable:

$$\mathbf{z}_{\text{bg}} \sim q(\mathbf{z}_{\text{bg}} \mid \mathbf{z}_r^{\text{appr}}, \mathbf{x}) \propto p(\mathbf{z}_{\text{bg}} \mid \mathbf{z}_r^{\text{appr}}) \, q_{\text{SPACE}}(\mathbf{z}_{\text{bg}} \mid \mathbf{x})$$

4: **return** background variable $\mathbf{z}_{\text{bg}}$

---

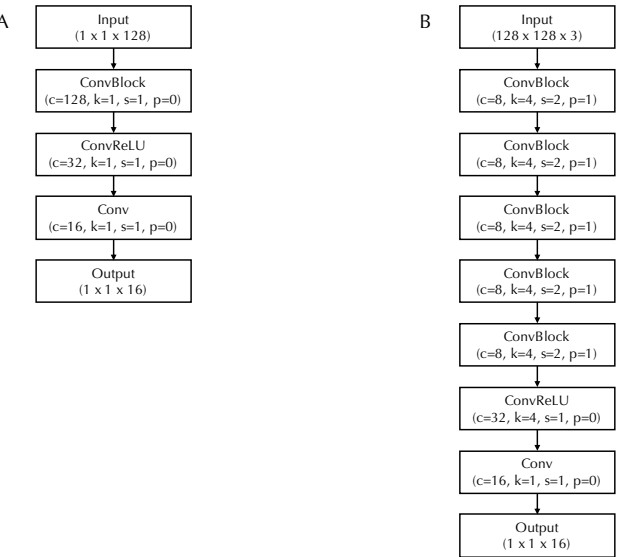

**Figure 10:** CNN architectures of *(A)* BgPri and *(B)* BgEnc used in our three-level GSGN.

## D.5 TRAINING DETAILS

We train GSGN on a single GPU, using Adam optimizer (Kingma & Ba, 2014) with a batch size of 64 and a learning rate of $3 \times 10^{-4}$, for up to 500 epochs. We use gradient clipping to ensure that the infinity norm of the gradient does not exceed $1.0$. The temperature for Gumbel-Softmax (Jang et al., 2016; Maddison et al., 2016) is exponentially annealed from $2.5$ to $0.5$ during the first 20 epochs. Similar to Slot Attention (Locatello et al., 2020), the learning rate is linearly increased from 0 to $3 \times 10^{-4}$ during the first 10 epochs, and exponentially decayed to half of its value every 100 epochs. We set $\sigma_{\text{fg}} = 0.3$ and $\sigma_{\text{bg}} = 0.1$. On the Compositional CLEVR dataset, $\sigma_{\text{fg}}^2$ has an initial value of $0.15^2$, and is linearly increased from $0.15^2$ to $0.3^2$ during epochs 20-40. Similar to SPACE (Lin et al., 2020b), the mixing weight $\hat{\mathbf{m}}_r$ is fixed at the start of training. On the 2D Shapes dataset, we fix $\hat{\mathbf{m}}_r = \mathbf{1 \times 10^{-5}}$ for 1 epoch, while on the Compositional CLEVR dataset, we fix $\hat{\mathbf{m}}_r = \mathbf{0.1}$ for 2 epochs. GSGN-9 is trained on two GPUs, each taking a batch size of 32, using the same schedule.

## E COMPARISON WITH SUPERPIXEL HIERARCHY

In this section, we qualitatively compare GSGN with a simple baseline called Superpixel Hierarchy (Wei et al., 2018). This algorithm hierarchically merges pixels into superpixels based on connectivity and color histograms, until the full image is merged into one superpixel. Using this hierarchy, the algorithm is able to produce segmentation results when given the desired number of superpixels (can be any number between one and the total number of pixels). It has been shown that Superpixel Hierarchy outperforms the widely used FH (Felzenszwalb & Huttenlocher, 2004) and SLIC (Achanta et al., 2012) methods and also some more recent methods.

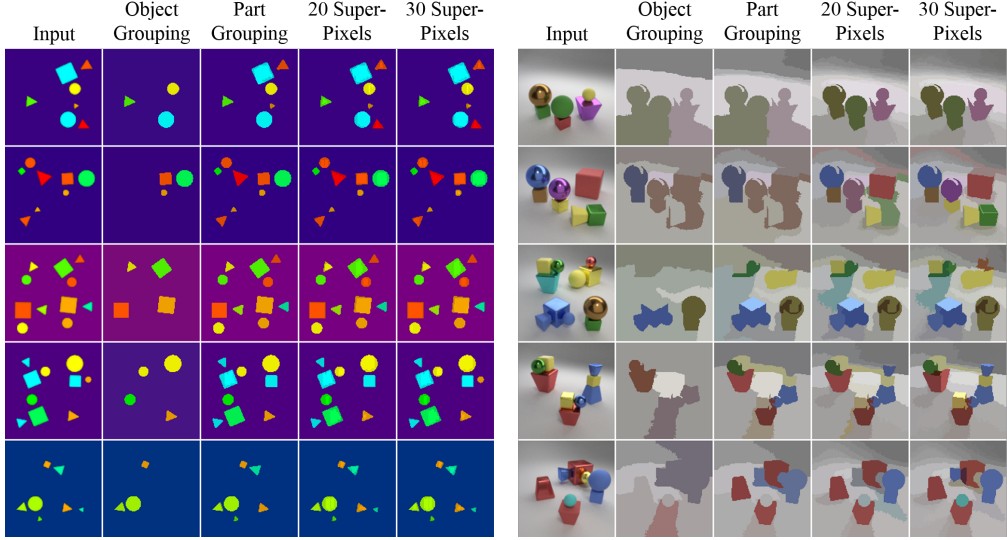

**Figure 11:** Segmentation results produced by Superpixel Hierarchy.

In principle, Superpixel Hierarchy (SH) should be able to perform object and part grouping, by assigning a superpixel to an object or a part. We show qualitative segmentation results on our datasets in Figure 11, using the publicly available code[2]. Here, we compute the mean color for each superpixel, and each pixel is drawn in the mean color of the superpixel to which it belongs. For object and part grouping, we provide SH with the groundtruth number of components (the number of objects/parts plus one for background). As can be seen, when doing object grouping, SH tends to merge some parts into the background. This seems to be caused by some characteristics of our datasets. Specifically, in the 2D Shapes dataset, the parts within an object do not touch each other, making it hard to merge them into one superpixel because superpixels are supposed to be 4-connected. In the Compositional CLEVR dataset, the light and dark regions of the background tend to be split into different superpixels due to dissimilarity in color histograms. For part grouping,

[2]https://github.com/semiquark1/boruvka-superpixel

SH sometimes merges tiny parts into the background on the 2D Shapes dataset, and splits metal parts that have specular highlights on the Compositional CLEVR dataset. When more superpixels are allowed, SH is generally able to distinguish foreground from background, but this kind of over-segmentation cannot serve our purpose of building the scene graph.

## F    RESULTS ON SEVEN-PART COMPOSITIONAL CLEVR DATASET

In this section, we show results on a slightly more complex version of the Compositional CLEVR dataset, in which we introduce four new parts and replace some of the objects. The new dataset contains ten types of objects composed from a total of seven parts. Three of the object types contain a single part, another three contain two parts, and the remaining four contain three parts. We use the same network architecture and auxiliary KL terms as those we used for the original Compositional CLEVR dataset. We find that adjusting the weight of the KL terms like in $\beta$-VAEs (Higgins et al., 2017) helps improve training stability. Specifically, we multiply the original and auxiliary KL terms of part-level presence and pose variables by a hyperparameter $\beta$. We keep $\beta = 1$ for the first 50 epochs, then linearly increase $\beta$ from 1 to 5 during the next 30 epochs, and finally keep $\beta = 1$ for the remaining epochs. We report qualitative results in Figure 12, and quantitative results in Table 10 and Table 11.

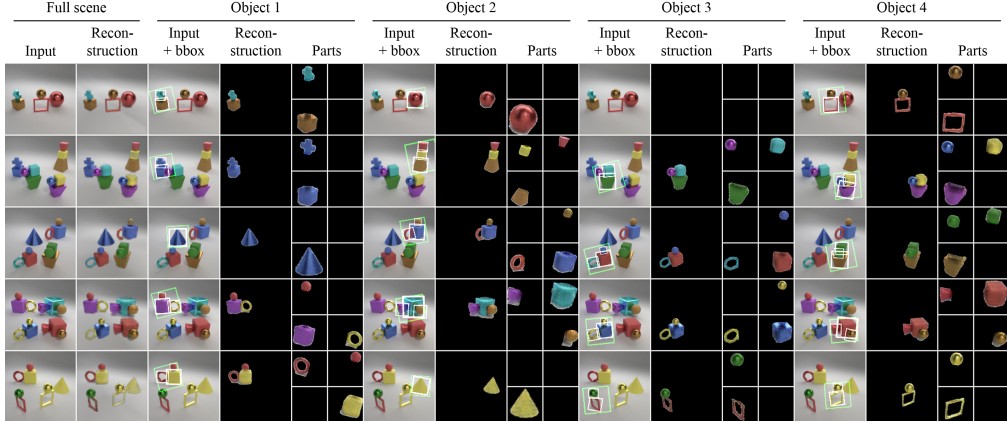

**Figure 12:** Visualization of inferred scene graphs on seven-part Compositional CLEVR dataset.

**Table 10:** Quantitative results on seven-part Compositional CLEVR dataset.

| Metric | ELBO | Object Count Accuracy | Object F1 Score | Part Count Accuracy | Part F1 Score |
|---|---|---|---|---|---|
| SPACE-O | 3.56 | 96.60% | 98.49% | — | — |
| SPACE-P | 3.55 | — | — | 89.69% | 92.10% |
| GSGN | 3.55 | 96.38% | 98.29% | 97.48% | 94.11% |

**Table 11:** Robustness to occlusion on seven-part Compositional CLEVR dataset.

| Min Visible Pixels Per Part | <100 | | 100~200 | | >200 | |
|---|---|---|---|---|---|---|
| Metric | Part Count Accuracy | Part Recall | Part Count Accuracy | Part Recall | Part Count Accuracy | Part Recall |
| SPACE-P | 0.00% | 81.05% | 82.78% | 94.50% | 93.39% | 90.25% |
| GSGN | 100.0% | 95.63% | 96.72% | 96.90% | 97.85% | 92.76% |

## G   ANALYSIS OF POSSIBLE SCENE GRAPHS

In this section, we provide a brief analysis of how variational the scene graphs can be on the Compositional CLEVR dataset.

Let us first investigate the possible scene graph structures (i.e., ignoring the difference caused by pose and appearance). There are 1-4 objects in a scene, each consisting of 1-3 parts. This leads to a total of $\sum_{n=1}^{4} \binom{n+3-1}{3-1} = 34$ possibilities.

Now we account for the pose variations of parts in the scene graph. Note that the relative position and scale of parts in each composite object are different. So the number of variations within an object is 1(single-part) + 3(two-part) + 4(three-part) = 8. This leads to a total of $\sum_{n=1}^{4} \binom{n+8-1}{8-1} = 494$ possibilities.

Finally, we account for the appearance variations. There are 7 colors and 2 materials, leading to 14 choices of appearance for each part. Combining with pose variations, we obtain $3 \times 14 + 3 \times 14^2 + 4 \times 14^3 = 11606$ possible subtrees for each object. Hence, the total number of possible scene graphs is $\sum_{n=1}^{4} \binom{n+11606-1}{11606-1} = 7.57 \times 10^{14}$.

We note that the above calculation still does not account for the pose variations of objects. In fact, object poses are sampled from continuous distributions as opposed to discrete ones, leading to infinite variations.

