# OpenReview forum: "Generative Scene Graph Networks"
_ICLR.cc/2021/Conference — ICLR 2021 Poster_

### Official Review · AnonReviewer3 · 2020-10-27
**Technically impressive, but is the complexity necessary?**

**Rating:** 6
**Confidence:** 4

**Review:**

**PAPER SUMMARY**

The paper presents a generative model for scenes that uses tree-structured latent variables to recursively decompose images into objects and parts, without any object or part supervision during training. The model is trained using variational inference. Experiments are performed on two new datasets (2D Shapes and Compositional CLEVR), demonstrating that the model is able to successfully uncover recursive scene/object/part decompositions in an unsupervised setting. The model is compared against prior work (SPACE) that performs non-hierarchical scene modeling.

**STRENGTHS**

- The paper presents a novel generative model that can infer tree-structured latent variables
- The method is technically impressive, and a clear improvement over the non-hierarchical modeling used in prior work
- The paper presents two new datasets (2D Shapes and Compositional CLEVR) for studying hierarchical scene decomposition. I hope these can be publicly released!

**WEAKNESSES**
- The method is quite complex, and though it is technically impressive I wish that it had been compared against very simple baselines
- No experiments on real-world data
- Unclear how the model will scale to wider and deeper trees
- Many implementation details are unclear

**SCALABILITY**

A big selling point of the proposed model is that it can model the hierarchy of scenes into objects and parts, and the tree-based formulation of the latent space used to achieve this is technically impressive. However, how well would the method scale to larger scenes? All experiments used a relatively small three-level hierarchy where all nodes have a fixed out-degree of 4; thus in all experiments the full tree has just 21 nodes total. How well can the method scale to much larger trees, with deeper hierarchies or wider out-degrees?

**PRIOR KNOWLEDGE**

The method requires choosing a tree depth and node out-degree as hyperparameters. To some extent, these hyperparameters encode very strong prior knowledge about the data being modeled (levels of hierarchy, and number of subparts within each part). In the experiments, these hyperparameters are perfectly matched to the synthetic datasets: you use a two-level tree with out-degree of four, and each image has between 1 and 4 objects, each of which is composed of between 1 and 3 parts. However in more complex real-world scenarios, you may not have such detailed knowledge of the world’s compositional structure. For this reason, I’m curious as to how the method would behave when the structural hyperparameters are mismatched to the underlying statistics of the dataset.

**SIMPLE BASELINES**

The model is evaluated on two synthetic datasets -- 2D Shapes, and Compositional CLEVR. How difficult is the scene graph inference problem on these datasets? I wish that the authors had compared to very simple baselines in order to give a sense for how difficult the problem really is.

For example, I suspect that a simple “handcrafted” baseline that oversegmented the image into superpixels, then recursively merged superpixels based on proximity and simple appearance features (e.g. color histograms) could produce very plausible scene graphs for these two datasets.

At a high level, the promise of an end-to-end learning-based approach compared to a “handcrafted” approach like the above is that a learning-based approach should require less tuning, should be more adaptable to new datasets and tasks, and should scale better to complex real-world datasets where the assumptions of the model don’t perfectly match the statistics of messy data from the real world. However in this case, I’m not convinced that this complex variational method would be any simpler to implement or tune, or even give better results than, a very simple “handcrafted” baseline like the above on these synthetic datasets. There are also no experiments to demonstrate its scalability to real-world datasets.

This leads to a pointed question: If someone wanted to infer scene graphs from images, why should they prefer your approach over a very simple “handcrafted” approach?

**MANY IMPLEMENTATION DETAILS UNCLEAR**

There are many implementation details that are unclear from the paper and supplementary material, without which reproducing the results are practically impossible. Even if some of these details do make sense as part of the main text, they should be specified more explicitly in the supplementary material. For example:

What is the generative process for the presence variables $z_v^{pres}$? This is not clear from Equations 3 or 4, nor the surrounding discussion. From Equation 9, the encoder predicts presence variables conditioned on image patches; but how can you predict the presence variables when sampling an image from scratch?

What is the dimension of the latent variables $z^{appr}_r?$

You say that the pose variables are Gaussian, but how is the pose parameterized in terms of relative location and scale?

From the discussion after Equation 6, the pose variables also include a depth map -- how are these depth maps represented and parameterized? Are they per-pixel Gaussians like the other pose variables?

How exactly are the transparency maps $\alpha_v$ computed from the depth maps? The text states that they are computed “by the softmax over negative depth values”, but this is hard to understand -- what are the sets of values being fed to softmax? If the depth map contains a per-pixel depth then a softmax over space wouldn’t make sense, since this would cause the transparency of an object to depend on its spatial size (since it would have more pixels competing in the softmax).

In Equation 9, the distributions of the latents $z_v^{pres}$, $z_v^{pose}$ are conditioned only on data from the parent node $z_{pa(v)}^{appr}$ and $z_{pa(v)}$. Thus all children of a node will have the same distribution for their pose and appearance. Is this correct? If so, how do you encourage the child nodes to cover all parts of the parent, and not collapse to a single part of the parent?

There are no details about any of the neural network architectures used to implement the model, nor any details about any training hyperparameters (e.g learning rates, training schedule, regularization strengths etc).

**SUMMARY**

The model is technically impressive, and a clear improvement over prior work. However on the whole I’m not sure whether the complexity of the method is actually necessary to solve the problem at hand; I wish that the authors had done a better job demonstrating the benefits of the proposed method over very simple baselines. There are also many implementation details that are very unclear, for which reason I fear that the paper as written is utterly unreproducible.

On the whole I lean slightly toward acceptance, but I hope the authors can address my concerns in their rebuttal.


**AFTER REBUTTAL**

The rebuttal largely addresses my concerns about implementation details.

I am pleased to see the additional experimental results provided by the authors; I think that these do improve the paper. I still feel that some well-tuned handcrafted approach could likely perform on-par with the results of the proposed method, but the comparison with [Wei et al] show that achieving such results is at least not trivial, which does help to better ground the complexity of the task. The additional experiments with a three-level hierarchy show a bit more evidence for scalability than provided in the original paper. While these extra experiments do strengthen the paper, I feel that they don't really address the core issue with the paper, which is whether there is any hope for the proposed method to scale to more complex and realistic datasets.

Overall I think that this is a reasonable paper and I still lean slightly toward acceptance, so I maintain my original rating of 6.

---

> ### Author Response · Authors · 2020-11-25
> **Response to Reviewer 3 (1/2)**
>
> Thank you for your insightful and very detailed feedback. We are encouraged that you find our method technically impressive and a clear improvement over prior work. We would like to address your concerns and answer your questions in the following.
>
> **IS THE COMPLEXITY NECESSARY?**
>
> The actual implementation is quite simple, although some complexity is needed for the mathematical formulation. To illustrate this, in the supplementary material we added Algorithm 1 and Algorithm 2 describing foreground inference (implementation of Equation 9), and Figure 9 depicting network architectures. From this, we expect that one will see how easy to implement this model. The main operations performed for decomposition at each level are just one feedforward pass through each of the three CNN submodules.
>
> > Compare to very simple baselines.
>
> Thanks for your suggestion. We added qualitative results from a simple baseline called Superpixel Hierarchy (Wei et al., 2018) in Section D. This algorithm hierarchically merges pixels into superpixels based on connectivity and color histograms, until the full image is merged into one superpixel. It can then produce segmentation results when given the desired number of superpixels. We evaluate its ability to perform object and part grouping by providing the groundtruth number of components (the number of objects/parts plus one for background) as the desired number of superpixels. We find that our datasets present some difficulties to this simple baseline. The 2D Shapes dataset contains very tiny shapes, which tend to be merged into the background. Also, the parts within an object do not touch each other, making it hard to merge them into a single 4-connected superpixel. In the Compositional CLEVR dataset, the shadows in the background and the specular highlights on metal materials cause color dissimilarity within components, leading to incorrect decomposition.
>
> > Why should someone prefer your approach over a very simple “handcrafted” approach?
>
> First, we showed that the simple baseline we considered was incompetent to infer the scene graph even when given the groundtruth number of objects and parts. Second, our model does more than scene graph inference:
>
> - It infers the appearance of each part, which enables the model to inpaint occluded parts (many examples in Figure 3)
> - It is able to composite new objects by reconfiguring the pose and appearance variables in an inferred scene graph (newly added Figure 4B)
> - It can perform unconditioned scene generation, which even SPACE cannot do (Figure 5)
> - The learned hierarchical representations can improve data efficiency in downstream tasks (Figure 6)
>
> We believe a model unifying the above capabilities is a meaningful contribution.
>
> **SCALABILITY & MISMATCHED STRUCTURAL HYPERPARAMETERS**
>
> We implemented GSGN-9, a three-level GSGN where each node has an out-degree of 9, resulting in a tree with 91 nodes. We evaluated its performance on the Compositional CLEVR dataset, and updated the results in Table 2 and Table 3. We observed a slight drop in performance when compared to GSGN. However, GSGN-9 is still much better than SPACE-P at identifying parts that have severe occlusion. The slightly worse part F1 score is caused by slightly inaccurate prediction of center positions. If we increase the error tolerance from 5 pixels to 6 pixles, then the part F1 score of GSGN-9 becomes 99.07%, which is comparable to other models.

---

> ### Author Response · Authors · 2020-11-25
> **Response to Reviewer 3 (2/2)**
>
> **IMPLEMENTATION DETAILS**
>
> We added implementation details to the supplementary material. We will also release the code and the Compositional CLEVR dataset upon publication.
>
> > What is the generative process for the presence variables $z_v^{pres}$?
>
> In Equation 4, the term $p(z_v^{pres} \!\mid\! z_{pa(v)}^{appr})$ suggests that during generation, the presence variable of node $v$ is conditioned on the appearance variable of its parent $pa(v)$. This makes sense because $z_{pa(v)}^{appr}$ is expected to summarize lower-level compositions. Note that this conditional distribution is a Bernoulli distribution, and as mentioned in the text right below Equation 4, its distribution parameter is learned by an MLP.
>
> > What is the dimension of the latent variables $z_r^{appr}$?
>
> On both datasets, the part-, object-, and scene-level appearance variables have dimensions 32, 64, and 128, respectively.
>
> > How is the pose parameterized in terms of relative location and scale?
>
> We clarified this in Section C.2. In short, the Gaussian variables are first converted to the desired range through nonlinear squashing functions, and then used to calculate the affine transformation matrix in the usual way required by the spatial transformer.
>
> > Depth maps and transparency maps.
>
> We clarified this in Section C.2. The pose variables are not per-pixel, but per-node variables. Specifically, we use a 1-dimensional depth variable for each node $v$ to represent the relative depth of $v$ with respect to its siblings (e.g., if $v$ is a part of object $u$, then the siblings of $v$ are the other parts of the same object $u$). This depth value is broadcast to all pixels that would belong to node $v$ if there were no occlusion. In regions of occlusion, each pixel will maintain multiple depth values. The transparency map is then computed by a per-pixel softmax over the negative depth values, to select for each pixel the entity that contains it and has the smallest depth.
>
> > In Equation 9, the distributions of the latents $z_v^{pres}$, $z_v^{pose}$ are conditioned only on data from the parent node $z_{pa(v)}^{appr}$ and $x_{pa(v)}$. Thus all children of a node will have the same distribution for their pose and appearance. Is this correct? If so, how do you encourage the child nodes to cover all parts of the parent, and not collapse to a single part of the parent?
>
> Good question. First, the distributions are not the same, because the network parameters are not shared among the children nodes of the same parent. In fact, the children nodes are inferred in parallel through a CNN. Second, similar to SPAIR and SPACE, we divide $x_{pa(v)}$ into grid cells. Each child node is associated with one cell and is encouraged to identify a part that is close to that cell. We made the above clear in the supplementary material by showing network architectures in Figure 9 and describing the parameterization of positions in Equation 15.
>
> > Details about neural network architectures and training hyperparameters.
>
> Thanks for the suggestion. We added Section C describing the implementation details.

---

### Official Review · AnonReviewer2 · 2020-10-28
**Not Sure If the Method is Significant Enough**

**Rating:** 4
**Confidence:** 3

**Review:**

=====Post-Rebuttal Comment=====

I thank the authors for the detailed response and the updated results. While my overall opinion of the work is slightly more positive post-rebuttal, I still maintain that this is a clear reject, primarily for the following reason:

- The technical contribution (adding a hierarchical layer to SPAIR and demonstrate the hierarchy can also be learned without supervision) is not significant enough to accept purely based on a "proof-of-concept" of a "new" direction.
- For incremental contributions, I expect the experimental results to be more convincing to be acceptable, a few points that I still really expect to see:
    - comparisons on harder datasets when one doesn't have to go into a specific metric to show an edge over a prior art that is targeting a different application
    - results on decomposition with more significant overlaps (especially in 2D) and on objects where part boundaries are harder to infer (actual 3D objects is still preferable)
    - object-level manipulation (row 3&4 in fig 4 did not match description) and latent space interpolation

I would also strongly encourage the authors to highlight the similarity and different between SPAIR and SPACE when introducing the latent code formulation.

=====Summary=====

This work proposes a method that can learn a three-level (scene, object, part) hierarchical representation of scenes in an unsupervised manner. A variational autoencoder is used here, where the encoder recursively infers the shape and pose of individual objects and parts, and the decoder recomposites the inferred parts are then recomposited to the inferred poses. The proposed method is evaluated on two datasets created for this paper. The results suggest that the proposed method can reliably breaks down scenes into meaningful objects and parts, and performs slightly better than another method designed for a slightly different task in terms of reconstruction quality, learned representation, and data efficiency for downstream tasks.

=====Strengths=====

- The motivation of the need for hierarchy is solid, and the solution proposed seems to me to be a reasonable way to impose some sort of hierarchy.
- The model seems to be well-tuned, utilizing appropriate training tricks and architectures.
- Good performance for the evaluations chosen in the paper.

=====Weaknesses=====

- *Very* inadequate attribution of ideas. I am not too familiar with the AIR line of work, but I think quite a few ideas can be traced back to prior works. It would be much better if the authors can, in addition to a brief one sentence mention in related works, add clear discussions for the inspirations of the main design choices.
- Missing discussions of relevant works that do: 1. unsupervised part decomposition e.g. “UCSG-NET - Unsupervised Discovering of Constructive Solid Geometry Tree”, "Bae-net: Branched autoencoder for shape co-segmentation", “CvxNet: Learnable Convex Decomposition”; 2. Learning hierarchical representations e.g. “StructureNet: Hierarchical Graph Networks for 3D Shape Generation”.
- I think the comparisons in this work are neither adequate nor fair. I am not convinced that the two toy dataset used here can prove the superiority of the method. The authors claim that “other works can’t work on our dataset”, but I think the burden of the proof is on the authors to show that their method is superior, even under a more specific setting. In other words, if the method is indeed “general” and can learn good decompositions, then I would expect it to perform better even under an slightly unfair setting i.e. comparing against metrics/datasets adopted in other works. Furthermore, the datasets used in this paper appears to be way too simple as compared to real world data. The authors argue that dataset with a single shape is easier, but I disagree: datasets like partnet contains much more complicated part structures, as well as joints between parts, than what is used here, even with only a single shape. (And it is pretty evident from the qualitative examples that the challenging part is decomposing objects into parts, not decomposing scenes into objects). Last but not least, I want to see more evidence that the proposed method is actually useful in real applications.
- The learned representation does not seem to be of very good quality, as seen in Figure 4.
- A lot of overclaims, to name a few: 1. “GSGN is a general framework for representing and inferring scene graphs of arbitrary depth”: I don’t think a model being able to work a toy setting with three levels will mean that the same framework can be used for more complex settings of arbitrary depth (as an analogy: MLP works for MNIST but not on ImageNet). If the framework can handle more general cases, then show it. 2. “Closely follow the rendering process in graphics engines”: I don’t think applying affine transformations and compositing alone is enough to warrant this claim, it is pretty clear that the learned representation lacks a good sense of “objectness”, as textures, lighting and etc. are all entangled together (evident in Fig 3). There does not seem to be a straightforward way to extend the method to truly parallel the 3D rendering process, neither. 3. “First deep generative model for unsupervised scene-graph discovery”: there are a lot of works that infer structures in an unsupervised way, I don’t think it’s fair to give a very narrow definition of “scene graph” and claim “the first”. 4. “GSGN has capture many predefined object types in the dataset”: I don’t think one can make this claim when there are only three primitives and ten types of objects…

=====Reasons for Score=====

Overall, I have the impression that this work is cherry picking a very specific setting where the proposed architecture works reasonably well. For a work making a quite big claim of being “the first deep generative model that learns …”, I would expect much more comprehensive evaluations than what is currently shown here. Furthermore, many ideas in this work are not attributed properly, making the novelty of the method quite unclear. From my limited knowledge of the direct predecessors of this work, I don’t think there is too much novelty in this paper. I would be more than willing to change my score if the authors can 1. Provide more comprehensive evaluations 2. State the novelty / discuss prior works more clearly. But for now, I tend to give a pretty clear reject.

=====Additional Comments & Questions=====

- I am not sure if translation & rotation alone is a good way to handle 2D renders of 3D objects, since any translation & rotation will result in change of perspectives and illumination e.g. rotating the bronze sphere in Figure 3, row 1 will make the specular highlight inaccurate and translating the blue cube in row 3 will make the top surface less visible. Could the authors justify why predicting translation/rotation make sense, when the perspective/illumination of the object already provides a really strong cue?
- Following previous point: would like to see examples of the same learned object being used in multiple scenes.
- The quality of the learned primitives, as seen in Figure 4, seems to be pretty underwhelming. If the aim of the work is discovering those primitives, would it make more sense to impose a stronger prior on the properties of the primitives?
- Table 1 & 2: why are all the ELBO terms the same? I would imagine them to be different, especially for SPACE-O/P, which, if I understand correctly, is a *completely different* model with different architecture and loss formulation?
- Table 2 & 3: why are the metrics so close between SPACE-P & GSCN in table 2 but so different in Table 3? Does that suggest unbalanced dataset?
- Still Table 2 & 3: why no comparison between SPACE-O & GSGN for object level occlusion?
- The paper claims that being able to handle background is a unique advantage as compared to other works, but the background used in the toy dataset is quite simple. Would like to see more complex examples.

---

> ### Author Response · Authors · 2020-11-25
> **Response to Reviewer 2 (1/2)**
>
> Thank you for your very detailed comments and suggestions for improvement. We would like to address your concerns and answer your questions in the following.
>
> > Unclear novelty and simple datasets.
>
> Please see our general response regarding this concern.
>
> > Very inadequate attribution of ideas.
>
> We improved the related work section as you suggested. We would like to mention that we did give credit to prior work when introducing our main design choices. For example, we cited relevant prior work when introducing parameter sharing and auxiliary KL loss.
>
> > Missing discussions of relevant works.
>
> Thanks for pointing to relevant works. We have added them to the related work section. In general, these works solve different problems than ours:
>
> - The first three methods take 3D voxels as input, and predict part geometry (i.e., occupancy). They only work for single objects, and cannot model appearance (color, material, etc). In contrast, our model takes 2D images of multi-object scenes as input, and infers both the decomposition and appearance of each part.
> - The first two methods have some additional assumptions. UCSG-NET assumes predefined primitive parts, such as box and sphere, while our model can learn to discover the primitive parts purely from data. Bae-net assumes the objects come from the same category, thus having similar part structures. In particular, all objects have the same number of parts. Our model can infer the number of parts, and does not require that all objects have similar structures.
> - In StructureNet, the part hierarchy is provided as input to the model, rather than inferred from the input.
>
> > Comparing against metrics/datasets adopted in other works.
>
> We have compared with the closest baselines, and we do not think it is fair or suitable to compare with other works. As we mentioned in the experiment section, "Previous work on hierarchical scene representations assumes single-object scenes (Kosiorek et al., 2019) and requires predefined or pre-segmented parts (Li et al., 2017; Huang et al.,2020)". We do not think it is fair to compare our model (without access to groundtruth parts) to (Li et al., 2017; Huang et al.,2020). Also, their datasets use voxels/point clouds as input, which is a different problem setting. In (Kosiorek et al., 2019), the main metric is unsupervised classification accuracy on MNIST, CIFAR10, and SVHN. These datasets do not seem suitable for evaluating interpretable part decomposition. And as we discussed above, the papers you mentioned solve different problems than ours, and also require different input formats.
>
> > The authors argue that dataset with a single shape is easier, but I disagree.
>
> We did not say that dataset with a single shape is easier. What we intended to argue is that methods that work on single-object scenes circumvent the challenge of grouping parts into objects, because they assume that all parts belong to the same object. Clearly this assumption cannot hold when there are multiple objects. Our model directly addresses this challenge by taking a top-down inference approach. This is in itself a contribution. If we took a bottom-up approach, then when objects are close, it would be very hard to determine which part should belong to which object. Also, the fact that SPACE-P cannot separate occluded parts indicates that the bottom-up approach would likely be sub-optimal. We believe that solving this part grouping challenge is orthogonal to learning complicated part structures of a single object.
>
> > Real applications.
>
> As suggested by Reviewer 4, we added Figure 4 showing an application of the learned scene graph in image manipulation. We demonstrated that by modifying the pose and appearance variables in a learned scene graph, one can individually manipulate the objects and parts in the scene. Also, by combining parts from multiple scenes, one can composite new objects that are never seen during training. While our model currently cannot deal with real-world data, it is likely that our inference module can be upgraded to incorporate future advancements that generalize scene decomposition models to more complex scenarios.
>
> > The learned representation does not seem to be of very good quality, as seen in Figure 4.
>
> We respectfully disagree. Figure 3 would be more appropriate for investigating the quality of learned representations. Figure 4 (Figure 5 in updated version) shows unconditioned generations. The latent variables are not inferred from a given image, but directly sampled from the prior, i.e., the root node is sampled from $\mathcal{N}(0,1)$, and others are sampled from conditional priors. Typically in VAEs, we cannot expect unconditioned generations to be of very good quality. However, it is clear from the figure that our model gives better object generations than SPACE. Also, our model can generate meaningful scenes while SPACE can only generate empty scenes.

---

> ### Author Response · Authors · 2020-11-25
> **Response to Reviewer 2 (2/2)**
>
> > A lot of overclaims, to name a few:
>
> Thanks for pointing this. We agree that the tone of some of our claims requires some adjustment. We made the adjustment as follows:
>
> > 1. “GSGN is a general framework for representing and inferring scene graphs of arbitrary depth”: I don’t think a model being able to work a toy setting with three levels will mean that the same framework can be used for more complex settings of arbitrary depth (as an analogy: MLP works for MNIST but not on ImageNet).
>
> We changed it to "While our formulation of GSGN allows representing and inferring scene graphs of arbitrary depth, in our experiments, we have only investigated the effectiveness of a three-level GSGN".
>
> > 2. “Closely follow the rendering process in graphics engines”: I don’t think applying affine transformations and compositing alone is enough to warrant this claim, it is pretty clear that the learned representation lacks a good sense of “objectness”, as textures, lighting and etc. are all entangled together.
>
> We changed it to "follow the recursive compositing process in graphics engines".
>
> > 3. “First deep generative model for unsupervised scene-graph discovery”: there are a lot of works that infer structures in an unsupervised way, I don’t think it’s fair to give a very narrow definition of “scene graph” and claim “the first”.
>
> We changed it to "first deep generative model for unsupervised scene-graph discovery from multi-object scenes without knowledge of individual parts".
>
> > 4. “GSGN has capture many predefined object types in the dataset”: I don’t think one can make this claim when there are only three primitives and ten types of objects
>
> We changed it to "GSGN has captured many predefined object types in the two datasets considered".
>
> **ADDITIONAL QUESTIONS**
>
> > Why predict translation/rotation, when the perspective/illumination of the object already provides a really strong cue?
>
> Translation/rotation are inherent components of the scene graph. They identify an image region as an object/part, allowing structured understanding of the scene. Our current model learns the 2D appearance of objects/parts in a single rendered image. It does not learn 3D appearance, which would require a set of input images from multiple viewpoints. Hence, we cannot explicitly or accurately model perspective/illumination.
>
> > Examples of the same learned object being used in multiple scenes.
>
> Please see row 3-4 of the newly added Figure 4A, where we replace an object of the current scene with an object from another scene. As we explained above, we cannot expect to model perspective/illumination changes from a single input image.
>
> > Table 1 & 2: why are all the ELBO terms the same?
>
> ELBO is usually dominated by the reconstruction error. Here, we normalized the ELBO by the number of pixels, which is common practice in VAE literature. A similar ELBO in this case indicates similar reconstruction quality. The precise ELBO values are different.
>
> > Table 2 & 3: why are the metrics so close between SPACE-P & GSGN in Table 2 but so different in Table 3? Does that suggest unbalanced dataset?
>
> Here are the statistics over the 12800 test images. We did not have direct control over the minimum number of pixels per part.
>
> | Min Visible Pixels Per Part | <100 | 100~200 | >200 |
> |:---------------------------:|:----:|:-------:|:----:|
> |    Number of Test Images    |  49  |  3772   | 8979 |
>
> > Table 2 & 3: why no comparison between SPACE-O & GSGN for object level occlusion?
>
> There is no significant difference in overall performance on object-level decomposition, so we thought it would be unnecessary.

---

### Official Review · AnonReviewer4 · 2020-10-28
**auto-encoder for object-part scene graphs in simple environments**

**Rating:** 6
**Confidence:** 3

**Review:**

Generative Scene Graph Networks (GSGN) is a variational auto-encoder with the intermediate representation being tree-like scene graphs. The leaf nodes stand for primitive parts and edges stand for poses to compose parts into objects recursively. The experiments are done in two image datasets of single color, simple shape 2D/3D objects: Multi-dSprites and CLEVR, and the model is able to discover objects without supervision.

**Strength**: I find the direction important, and the method well established in a variational inference framework with graphics-inspired designs. Experiment numbers look good in general.

**Weakness**: Perhaps my biggest concern is that the current datasets are a bit weak. First, objects are too simple (single color, simple shape), so we are not sure if GSGN can work with more realistic visual domains where objects have more complex 3D structure or texture. Second, the object/primitive decomposition is slightly weird to me. I would expect hierarchal structure like an object being human body, and parts being legs, arms, head and so on. But in this work a "part" is a single-color object, and an “object” is a bunch of single-color adjacent objects. Based on these two points, I believe the paper will be much stronger with experiments on more complex objects like humans or tables or

Another concern is the application of learned scene graph. The paper only shows unconditional sampling, but not really how to use the learned scene graph. For example, I'd expect scene graphs to be used for image manipulation, as one can change part of the object (shape, color, pose) without changing the rest. Showing the learned scene graph is useful for any downstream tasks can be a great plus for the current work.

Finally, I wonder how variational the learned scene graphs can be, as the objects in the datasets are fairly simple and the learning might be easy. I'd be happy to see some analysis but this is not my main concern.

---------

After rebuttal: I'm glad they added some experiments and analysis I wanted to see, so I raise the score to 6. As the authors said, the paper is a proof-of-concept of unsupervised hierarchal scene graph learning, and the rebuttal to some degree reassured me. For example, modeling a cube on top of another top as two parts of an object (which was weird to me: why not each cube as an object?) helps edit tasks where the top cube is enlarged but the "on top" relation is maintained. The downstream representation transfer also makes sense. Of course experiments are still toy from computer vision perspective, but I'm now okay with acceptance.

---

> ### Author Response · Authors · 2020-11-25
> **Response to Reviewer 4**
>
> We appreciate your constructive comments. We would like to address your concerns below.
>
> > Objects are too simple (single color, simple shape).
>
> Please see our general response regarding this concern. We also made a slightly more complex version of the Compositional CLEVR dataset with four new parts. We obtained similar results on this new dataset (Section E).
>
> > The object/primitive decomposition is slightly weird.
>
> We agree that the compositional objects in our datasets do not look realistic. However, given that state-of-the-art unsupervised object-level decomposition models can only deal with simple shapes, we think it is reasonable to use these shapes as primitives. Also, some simple real-world objects like dumbbells and hair dryers have similar structures as those in our datasets, albeit with more complex shapes as primitives.
>
> > Application of learned scene graphs (e.g., image manipulation).
>
> Thanks for your suggestion. We added image manipulation results in Figure 4. We showed that one can individually modify the position and scale of objects and parts. In addition, we demonstrated that by reconfiguring the appearance variables, one can add objects from other scenes, and generate novel objects by compositing parts from multiple scenes.
>
> > Analysis of how variational the learned scene graphs can be.
>
> We added some analysis in Section F, regarding the number of possible scene graphs.

---

### Official Review · AnonReviewer1 · 2020-10-28
**A convincing proof-of-concept solution for a difficult new task**

**Rating:** 6
**Confidence:** 4

**Review:**

== Update ==

Thank you for your response and clarifications. I have left my score as is.

== Original Review ==

The paper presents an unsupervised method for inferring scene graphs from images. Building upon
scene-attention methods such as AIR and SPACE, it hierarchically decomposes a scene into objects and
those objects into parts, giving rise to a tree structure. It is shown that this model successfully
recovers the hierarchies underlying the data on two newly proposed hierarchical variants of the
Sprites and CLEVR datasets.

Strengths:
 1. The paper is well written, and despite the considerable complexity of the method, its
    presentation is relatively easy to follow.
 2. The task of interest is well-defined, and has clearly been effectivly solved on the datasets
    considered. Both quantitative and qualitative evaluations make it very clear that the model has
    learned to infer the correct scene graphs as desired. Its ability to infer the appearance of
    occluded parts is especially impressive.
 3. While there are no direct competitors on this newly defined task, the paper does a decent job of
    comparing to the closest available baseline, showing how the additional structure can be
    beneficial.

Weaknesses:
 1. One may argue that the datasets have been deliberately constructed to showcase the model. While
    that is probably true, I think this is a valid approach given the novel nature of the task and
    the lack of supervision. Despite the clearly helpful structure (limited number of objects and
    parts), the datasets still appear sufficiently challenging.
 2. In the experiments, scene graphs are limited to trees of height 2 and degree 4. This is a
    significant constraint, however, each additional level of hierarchy introduces ambiguities and
    makes it harder to learn the graph in an unsupervised manner. More complicated structures would
    likely require supervision.
 3. As far as I can tell, the object types were chosen once to generate the datasets, and then kept
    fixed across the experiments. Reporting results for multiple different datasets with randomly
    chosen object types would be somewhat more convincing.
 4. As is common for unsupervised scene models, the proposed method likely only works on synthetic
    images in its current state. However, due to the additional structural assumptions on the data,
    it seems especially challenging to find suitable real-world use-cases.

Overall, the paper presents an effective new method for the task it sets out to solve. While it is
questionable how it would work on real-world data, I believe the paper is of sufficient interest as
a proof of concept, and am therefore leaning towards acceptance.

Questions:
 1. It is stated that auxiliary KL terms are added, with the sparseness constraint on $z^{pres}$
    being one of them. But is not clear if there are others. This would be important to know in
    order to evaluate how strong the model's inductive biases are.
 2. The downstream task used for Fig. 5 is not clear to me. If the number of parts is computed for
    each object, and these numbers are then summed, isn't the result equal to the total number of
    parts in the scene, which SPACE-P can also infer? If only distinct parts are counted,
    how is equality of parts defined on the dataset?

---

> ### Author Response · Authors · 2020-11-25
> **Response to Reviewer 1**
>
> Thank you for your positive and thoughtful comments. We are encouraged that you find our paper well written and the inference of occluded parts impressive. We would like to address your concerns and answer your questions in the following.
>
> > Deliberately constructed datasets with limited number of objects and parts.
>
> We appreciate that you think this approach is valid and the datasets are sufficiently challenging. While the number of objects and parts are limited, we designed each object to have a different structure in terms of the relative pose of its parts. This helps cover a wide range of scene graphs, of which we added an analysis in Section F.
>
> > Limited tree height and width.
>
> We implemented GSGN-9, a three-level GSGN where each node has an out-degree of 9. We evaluated its performance on the Compositional CLEVR dataset, and updated the results in Table 2 and Table 3. GSGN-9 is still much better than SPACE-P at identifying parts that have severe occlusion. The slightly worse part F1 score is caused by slightly inaccurate prediction of center positions. If we increase the error tolerance from 5 pixels to 6 pixles, then the part F1 score of GSGN-9 becomes 99.07%, which is comparable to other models. While we did not evaluate GSGN-9 on more complex data, our results demonstrate the effectiveness when structural hyperparameters are mismatched to data (suggested by Reviewer 3), and show some potential for scalability.
>
> > Reporting results for multiple different datasets.
>
> Thanks for the suggestion. We made a slightly more complex version of the Compositional CLEVR dataset with four new parts. Results are similar, and we added them in Section E. Although the object types were not randomly chosen, we believe this new result is still meaningful.
>
> > Real-world use cases.
>
> We agree that our model currently cannot deal with real-world data. However, because we use scene decomposition models as an inference module, our model is likely to benefit from future advancements that generalize scene decomposition models to more complex scenarios.
>
> **ADDITIONAL QUESTIONS**
>
> > Auxiliary KL terms.
>
> We added description of auxiliary KL terms in Section C.3. We note that the prior distributions in these terms were not chosen to match the actual distributions in the datasets. Also, we used mostly the same terms on both of our datasets.
>
> > How is the number of parts computed in the downstream task?
>
> We clarified this in our description. Only distinct parts are counted within each object. Two parts are considered the same if they have the same shape, regardless of their pose, color, and material.

---

### Author Response · Authors · 2020-11-25
**To All Reviewers**

We thank all reviewers for their time and insightful feedback. In this general response, we would like to address the main concern about the simplicity of our datasets.

We understand that some prior work on part decomposition has been applied to real-world datasets such as ShapeNet and PartNet. However, these works have three significant differences from ours. First, most of the works require some kind of supervision while ours is fully unsupervised. This usually includes 3D supervision (requiring voxels or point clouds as input) and part-level supervision (providing pre-segmented parts or decomposing into predefined parts like boxes and spheres). Second, they typically focus on geometry (e.g., part occupancy) without learning to represent appearance (e.g., color, material) simultaneously. Third, most of these methods can do either inference or generation, but not both. Lastly, most of these models do not support the controllability of the representation to generate novel out-of-distribution scenes.

A recent line of work on unsupervised object-centric representation learning aims to eliminate the need for supervision in structured scene understanding. These methods learn **a holistic generative model capable of decomposing scenes into objects, learning appearance representations for each object, and generating novel scenes via controllable composition of object representations---all without supervision and in an end-to-end trainable way.** We believe such unsupervised and holistic models are more desirable, albeit more challenging to learn.

Although it is a promising approach with great potential in the long run, the unsupervised object-centric representation learning approach has a more ambitious goal. The methods are in its very infancy, and thus currently have some difficulty dealing with real-world data; the most complex datasets used in state-of-the-art models are evaluated on CLEVR-like datasets. Some representatives include IODINE (ICML 2019), SPACE (ICLR 2020), and Slot Attention (NeurIPS 2020). However, we believe future advancements will likely generalize them to more complex data. Thus, **as pointed by Reviewer 1, we hope the reviewers to see our paper as a proof-of-concept paper about a challenging but promising direction.** Accordingly, we will make our claim about proof-of-concept clearer.

In this paper, we take a step forward in this specific line of research by further decomposing objects into parts. Our model is also unsupervised and holistic, in the sense that it can infer part-object structures, learn appearance representations, perform image manipulation (newly added Figure 4), and generate objects and scenes. **We are not aware of existing part decomposition methods that can do all of the above in a single model without supervision.**

While we agree that it would have been better if we made our model work on complex natural images, we believe that our compositional CLEVR dataset, where an object is composed of several CLEVR-like shapes, is still **significantly more complex than the datasets used in previous works in the same line of research.** We showed that the severe occlusion among parts presents some difficulty to SPACE (Table 3). We believe that by introducing hierarchical structures and demonstrating effectiveness on this more challenging dataset, our model makes significant progress in the line of unsupervised object-centric representation learning.

During the rebuttal period, we made a slightly more complex version of the Compositional CLEVR dataset, by introducing four new parts. We obtained similar results on this new dataset (Section E). We acknowledge that this is still simple compared to real-world data, and we expect that by upgrading the inference module, our model would benefit from future advancements that can decompose more complex scenes.

---

### Decision · Program_Chairs · 2021-01-07
**Final Decision**

**Decision:**

Accept (Poster)

**Comment:**

This submission generated significant discussion between the reviewers; three of them ended up on the "accept" side, but one remained firmly in the "reject" camp.

The main strength of the paper is that it tackles a very hard problem: learning an unsupervised generative model (and accompanying inference model) of scene graph structures given only image data. As one reviewer mentioned, it is remarkable that the authors were able to get their system to work at all, given the seeming intractability of this problem. The work builds upon a clear line of prior work in this area, and the type of data on which it is evaluated ("toy" synthetic datasets a la CLEVR) is consistent with prior art.

Multiple reviewers brought up the "toy" nature of the dataset as a drawback to the paper, but most agreed that this is not reason to reject the paper. Rather, the paper demonstrates a convincing proof of concept that this kind of model can be built, and improvements in the elements out of which the model is composed (generative and inference networks) should improve its applicability to real-world data.

Another question mark raised by multiple reviewers: could a simpler, handcrafted inference procedure work just as well or better? The authors included a new experiment against a hand-coded heuristic in their rebuttal, and their method outperforms it. One reviewer noted that more careful tuning might make a heuristic perform as well as the proposed method, but it is still clear that it is not trivial to get a hand-coded solution to perform well (even for this "toy" data). Another reviewer pointed out that this is one of the main attractions of variational inference methods: the ability to specific knowledge as simple generative priors rather than complex bottom-up inference procedures.

One reviewer, R1, remains negative about the paper. His (it is a he; I know this reviewer) main concern is that the scene graphs used are shallow and have a simple structure, and thus (a) it's not clear what value they add, (b) a simple postprocess could reconstruct them, assuming the individual object parts could be detected, and (c) it's not clear whether the method would generalize to deeper/more complex hierarchies. He believes this calls into question the validity of the entire method.

I am sympathetic to this argument, but I think setting the bar this high may prevent progress in this field. For point (a), the authors included an image-manipulation application in their rebuttal--again, a proof of concept, not a directly useful tool. For point (b), the authors did compare against a hand-coded inference baseline and achieved better results, so while this may be possible, it is probably not as easy as the reviewer suggests. (c) remains an open question, to me. But even if this method as presented cannot generalize to more complex scene graphs, it likely paves the way for future work that can.